# COVID-19 lockdowns and demographically-relevant Google Trends: A cross-national analysis

**Lawrence M. Berger[1], Giulia Ferrari[2], Marion Leturcq[2], Lidia Panico[2], Anne Solaz[2]***

**1** University of Wisconsin-Madison, Madison, Wisconsin, United States of America, **2** Institut national d'études démographiques (Ined), Paris, France

\* solaz@ined.fr

**Data Availability Statement:** The data underlying the results presented in the study are available from Google search engine. We have also made the minimal data set underlying the results, as well

## Abstract

The spread of COVID-19 and resulting local and national lockdowns have a host of potential consequences for demographic trends. While impacts on mortality and, to some extent, short-term migration flows are beginning to be documented, it is too early to measure actual consequences for family demography. To gain insight into potential future consequences of the lockdown for family demography, we use cross-national Google Trends search data to explore whether trends in searches for words related to fertility, relationship formation, and relationship dissolution changed following lockdowns compared to average, pre-lockdown levels in Europe and the United States. Because lockdowns were not widely anticipated or simultaneous in timing or intensity, we exploit variability over time and between countries (and U.S. states). We use a panel event-study design and difference-in-differences methods, and account for seasonal trends and average country-level (or state-level) differences in searches. We find statistically significant impacts of lockdown timing on changes in searches for terms such as wedding and those related to condom use, emergency contraception, pregnancy tests, and abortion, but little evidence of changes in searches related to fertility. Impacts for union formation and dissolution tended to only be statistically significant at the start of a lockdown with a return to average-levels about 2 to 3 months after lockdown initiation, particularly in Europe. Compared to Europe, returns to average search levels were less evident for the U.S., even 2 to 3 months after lockdowns were introduced. This may be due to the fact, in the U.S., health and social policy responses were less demarcated than in Europe, such that economic uncertainty was likely of larger magnitude. Such pandemic-related economic uncertainty may therefore have the potential to slightly increase already existing polarization in family formation behaviours in the U.S. Alongside contributing to the wider literature on economic uncertainty and family behaviors, this paper also proposes strategies for efficient use of Google Trends data, such as making relative comparisons and testing sensitivity to outliers, and provides a template and cautions for their use in demographic research when actual demographic trends data are not yet available.

as our program files (Stata do files), available at dx.
doi.org/10.17504/protocols.io.br3xm8pn.

**Funding:** The authors received no specific funding
for this work.

**Competing interests:** The authors have declared
that no competing interests exist.

## 1. Introduction

The spread of COVID-19 and the resulting local and national lockdowns imposed in most developed countries in 2020 may have a range of potential consequences for demographic trends. The most direct and visible consequences, which have already begun to be documented, include increases in overall mortality rates [1], potential changes in the distribution of deaths by age and cause [2], and disruptions in global and local migratory [3, 4]. It is still too early to measure the consequences of the pandemic for family demography in areas such as fertility, marriage, separation, and divorce, however, because trends therein are linked to longer-term emotional, biological, legal, and financial processes. Thus, current scholarship has been able only to speculate about possible consequences in these domains, although insights can be gleaned from relevant historical events, for example the 2008 economic recession [5] and the 1918 Spanish Flu pandemic [6–8]. Surveys performed during the lockdown have produced useful information about fertility intentions and changes therein as a result of the pandemic [9], but they cannot yet provide evidence about fertility realization. To gain further insight into potential future consequences of the lockdown for family demography, we use cross-national Google Trends search data to explore whether trends in searches for demographic-related terms changed during lockdowns in Europe and the United States.

There are many reasons to hypothesize that the pandemic and related lockdowns will have an impact on future family demographic trends [6, 10] in developed countries. Yet, the *a priori* direction of effect is ambiguous in many domains. For example, lockdowns may be associated with increased relationship quality as couples share more time together or with decreased relationship quality if they are experiencing considerable stress. Moreover, union dissolution through breakup or divorce may be constrained by limited economic opportunities, flexibility to change housing units, and economic uncertainty at the same time that increased stress, anxiety and, for co-resident couples, time and space to decompress from one another may be associated with increased union dissolution. Indeed, recent research suggests that the pandemic has been associated with increases in domestic violence (see [11–13]). In addition, lockdowns have forced a new balance between work and family responsibilities for many couples, especially parents of young children, and these changing arrangements have not necessary been shared equally between partners [14, 15], which could be another potential source of relationship stress.

Lockdowns may also be associated with increased (unplanned) fertility if sexual activity increases, perhaps because couples decide to move in together during (or because of) the lockdown. Alternatively, they may decrease fertility if economic uncertainty leads to greater precautions against pregnancy or if (particularly non-coresident) couples decrease their sexual activity or postpone cohabitation or marriage. Of particular note, however, we do not expect increased mortality as a result of the pandemic to have much of an impact on family formation or fertility simply because COVID-related deaths have been highly concentrated among the elderly [16, 17]. Neither increased mortality among reproductive age adults (including pregnant women) nor COVID-related (positive or negative) changes in lost pregnancies or stillbirths are likely to be large enough in magnitude to drive changes in overall fertility rates. Rather, the pandemic is more likely to have indirect influences on fertility, family formation, and family dissolution through changes in economic and socioemotional wellbeing that may result in changes in intentions and behaviors regarding dating, sexual activity, contraception use, couple formation, cohabitation, marriage, fertility, separation/divorce, and so forth, including the *timing* thereof [10].

The adverse economic effects of the pandemic have already been widely documented, as has variation in countries' policy responses intended to buffer them [18]. A large and

longstanding literature indicates that economic shocks and economic uncertainty are associated with subsequent family demographic trends. Research on links between economic uncertainty and fertility, for example, suggests that financial security and the ability to plan for the future has a positive impact on fertility [19–21], whereas adverse economic conditions and shocks are associated with delays and declines in fertility [22–26]. Likewise, lower levels of consumer confidence are associated with lower fertility rates, with the largest association *vis-à-vis* first births and a smaller association for subsequent births [27]. Luppi et al.'s [9] survey of five European countries found evidence that the pandemic resulted in young adults reducing their fertility intentions. There is also evidence that adverse economic shocks and downturns are associated with delays or declines in entry into cohabitation and marriage [28, 29].

Turning to union dissolution, theoretical models linking poor macroeconomic conditions with separation and divorce suggest competing hypotheses: poor economic conditions may increase risk of union dissolution as a result of increased financial and other stresses experienced by individuals and families; at the same time, union dissolution becomes relatively more expensive in a context of poor economic conditions during which pooling resources and realizing economies of scale may be particularly important [30, 31]. The weight of the empirical evidence from the United States and Europe, however, tends to support the latter hypothesis—that union dissolution declines during a poor economy [26, 30–35].

Beyond their economic impacts, the widespread lockdowns used to combat virus transmission have mechanically posed barriers to forming and maintaining social relationships outside one's immediate household and have likely affected individuals' physical and mental wellbeing, as well as their ability to plan for the future. Uncertainty about the length and progression of the pandemic (and associated governmental responses, including lockdowns), as well as about current and future economic and social opportunities appears to be driving increased stress and anxiety [36–38], which may, in turn, affect decision-making processes and demographic behaviors.

Rather than having uniform impacts, the economic consequences of the pandemic and associated lockdowns, spanning rising unemployment, declining earnings, and contraction of local and national economies, are likely to differentially impact individuals' and families' financial wellbeing as a function of the generosity and reach of public transfers afforded by their country's public policy response. As such, we expect considerable heterogeneity across countries—and, in particular, between Europe and the United States—in the degree to which individuals and families experience economic uncertainty and hardship. This heterogeneity is also likely to be reflected in differences in the magnitude and direction of patterns of relationship formation, quality, and dissolution, as well as fertility plans, as a consequence of heterogeneity in lockdowns, economic conditions, and policy responses by country [9]. Specifically, we hypothesize that lockdowns will lead to fewer demographic consequences in countries that better managed the economic fallout from the pandemic and provided a stronger financial safety net. Generally speaking, these were the European nations. In contrast, the U.S. and, to a somewhat lesser extent, U.K., provided weaker and less holistic responses. Consistent with this notion, scholars have observed diverging trends over time such that perceived threat from the pandemic, which started quite high in many European countries (Italy, Germany), has decreased considerably in Europe over time, while it has increased in the U.S. and U.K.; also, confidence in the ability of public institutions to combat the pandemic is particularly low in the latter countries [39].

Because observational data on demographic behaviors are available only with a considerable lag, we examine potential demographic consequences of the COVID-19 pandemic for future family and fertility trends by assessing trends in family demography-relevant Google search terms as a function of national lockdowns in Europe and state lockdowns in the United

States. Assessing trends in online searches is informative to understanding population-level worries, preoccupations, and future plans with respect to union formation, fertility, and union dissolution; moreover, recent research indicates that Google search trends are useful for predicting subsequent economic [40, 41] and demographic trends [42, 43].

Thus, the relative commonality of web searches for a range of family demography-relevant terms, and variation therein by country, is likely to provide insight into future trends in these domains. Because lockdowns were not anticipated, nor identical in timing or intensity across countries (and sometimes within single countries), we leverage variability in lockdown timing within and between countries to identify effects of lockdowns on demography-relevant internet searches with potential implications for subsequent demographic trends.

## 2. Data

The Google search engine is by far the most used internet search tool [44]: Google searches represent 91.4% of total internet searches in December 2020 and; this share is very stable over time. Google Trends data are available for words or phrases searched over a selected time period in a selected country or subnational area, such as U.S. states. As such, two main decisions were crucial to our study: the list of search terms to be chosen and the list of countries to be analyzed. With respect to search terms, we first selected a list of macro-topics related to demographically-relevant constructs that may be influenced by lockdown policies. These focused on sexual behaviors, contraceptive use, pregnancy termination and fertility; dating and couple relationships; marriage; and separation and divorce. We then identified a series of search terms for each topic, based on brainstorming with experts in a particular domain (including our teams members and colleagues spanning population studies disciplines). For example, terms such as contraception, abortion, pregnancy test, family planning, child plan, and additional/other child were identified as indicators of sexual behaviors, contraceptive use, pregnancy termination and fertility. These terms were then translated into the primary language(s) spoken in each analysis country (see S1 Table for the full list of terms and languages searched in each country). We used search terms rather that search topics, which also provided by Google Trends, to facilitate our ability to check the adequacy of English words selected and our translations thereof to capture the intended construct. We chose to use experts within each observed country to translate the search terms rather than relying on Google translations to ensure a better match of the translations with our intended meaning and to consider a larger and more relevant set of possible country-specific search terms. As some terms had country- or language-specific meanings that were inconsistent with our intent, or for which direct translation did not necessarily convey similar meaning (for example, "*avocat*" in French translates to both "lawyer" and "avocado"), translations focused on achieving consistent cultural, linguistic, and in-country meaning rather than direct translation of specific words. We further culled the most frequent related queries in Google searches to ensure that we were correctly identifying the construct for which we were looking. On this basis, we sometimes dropped words that were too ambiguous such as "date" in English or "sitio de citas" in Spanish because many such searches referred to other types of rendezvous (to a doctor for instance) rather than only to intimate ones.

We selected a representative group of European unilingual countries that were among those most affected by the COVID-19 epidemic and that imposed at least partial lockdowns. An additional methodological constraint was that at least one coauthor or colleague could understand the language in order to check the adequacy of the search terms. These included Austria, France, Germany, Italy, Spain, and the United Kingdom (as a whole). We also included the United States, for which there was considerable variation in both lockdown and

social welfare response at the state level, but which also generally adopted less strict lockdowns and a less generous social welfare response (see, e.g., [45, 46]) than the European nations. Indeed, evidence suggests considerable declines in family well-being in the U.S. in the wake of the pandemic [47, 48]. Although comparable Europe-wide data are not yet available, recent evidence from Germany also suggests declines in well-being, particularly for adults with children [49]. In order to observe the potential impact of lockdown policies on search trends, we also collected country- and state-specific lockdown dates (See S2 Table) as well as weekly data on COVID-19 cases and deaths from the Johns Hopkins University Center for Systems Science and Engineering [50]. According to World Bank data, all of the countries we observe have a very large prevalence of internet users: in 2019, this included 74% of those in Italy, 83% in France, 87% in United states, 88% in Austria and Germany, 91% in Spain, and 93% in United Kingdom [51].

We performed data extraction via the Google Trends Application Programming Interface (API) using R software. We set the extraction date to be weekly and extracted data from the first week of January 2016 through the last week of June 2020. This allowed for a sufficiently long observation window prior to the start of the pandemic to account for possible seasonal variation when observing search trends in the wake of the pandemic. We coded each week as starting on Sunday. For instance, in 2020, week 1 starts on Sunday January5[th]. When there is a 53rd Sunday in the year, both the 52nd and 53rd were coded as 52, such that the last week of the year is always coded 52.

Our analytical sample includes 234 weekly observations for each country (state) over the four-and-a-half-year period between January 2016 and June 2020. Pooling the European countries resulted in a sample of 1,404 weekly observations of the 6 sample countries. The U.S. sample for our panel event-study analyses (see below) includes 10,062 weekly observations of the 42 states, plus Washington, D.C., which implemented a lockdown; the sample for our difference-in-differences analyses includes 11,934 weekly observations of the 50 states and Washington, D.C. We included the 8 states that did not implement a lockdown in the difference-in-differences analyses as control states. These states could not be used in the panel event-study analyses since no lockdown date is observed for them. Note that, because Google Trends reports a null value for search terms that do not meet a minimum threshold within a state and week, we exclude some U.S. states from our analyses of particular terms if the search threshold was not met for all observed weeks.

Google Trends does not directly report the number of searches for a given search term. Instead, it provides an *index* of search activity reflecting searches for a term in a given time period relative to overall search volumes. The index is based on the fraction of queries that include the term in question in the chosen geographic unit at a particular time, relative to the total number of all Google queries in the same geographic unit at that time (for a thorough description, see [52]). When the denominator (total volume of Google searches) is roughly constant over time, variations in the index reflect changes in the actual number of searches for a given term. If the denominator changes, however, such that the global search volume in a given geographical area decreases or increases (a likely event in the case of a shock such as the introduction of lockdowns, see further discussion below), this index must be interpreted as the relative change in searches for a term proportional to the overall search volume. Therefore, if the overall search volume is increasing, a decrease in the index does not necessarily reflect a decrease in the number of actual searches for that term, rather that searches for that term represent a smaller proportion of all searches. The number of searches could, in fact, be stable while the total number of Google searches increases. Thus, the index describes only *relative* searches for a given term.

In addition, google does not provide the exact value of this index, but rather transforms it to a value ranging from 0 to 100. The algorithm sets the maximum value of the index of search

activities in the chosen geographic unit over a given time period to be 100. That is, a value of 10 means that the fraction of queries for a search term among all queries was one tenth of the maximum searches for that term at a given time (in our case, week) relative to the maximum searches for the term in the geographic unit over the full observation period (in our case, 234 weeks). Because the maximum value is area-specific, the time-series of *indexes* cannot be directly compared across countries or U.S. states.

In our analyses, Google Trends data on searches for each term were modeled as the ratio of the index of relative searches for the term (in a country or U.S. state in a given week) relative to the average index of search activity for that term in the country or state in any week over the observation period. That is, we transformed the raw data extracted from Google Trends into relative weekly data that are comparable across countries or states by dividing the index of search activity provided by Google Trends for each search term in a given week by the average index of search activity for that term in all weeks of the observation period using the following strategy.

Let $S_{cyw}$ be the index of relative search interest for a search term for week $w$ of year $y$ in country $c$ (or state $c$ in the U.S. case). $S_{cyw}$ is not observed. Let $S_c^m$ be the maximum index of weekly relative searches for the search term over the period. Google Trends provides the relative search index $R_{cyw} = S_{cyw}/S_c^m$ only. The raw Google Trends data are not suitable for cross-country comparison because these data provide only the index of search activity related to a maximum $S_c^m$, which is country specific. To transform the raw series into country-comparable series, we computed $\bar{R}_c$, the average index of country-specific searches $R_{cyw}$ over the period. We then construct a transformed series $\tilde{R}_{cyw}$ defined as:

$$\tilde{R}_{cyw} = \frac{R_{cyw}}{\bar{R}_c}$$

It can be shown that $\tilde{R}_{cyw} = \frac{S_{cyw}}{\bar{S}_c}$, where $\bar{S}_c$ is the average index of relative searches for a term in country $c$ over the January 2016-June 2020 period. Our transformed series $\tilde{R}_{cyw}$ represents the index of searches for the term (word or group of words) in week $w$ of year $y$ in country (or state) $c$, relative to the average weekly index of search activity for that term in the country (or state), over the period. The transformed series is comparable across countries and states and over time because variation in $\tilde{R}_{cyw}$ can be interpreted relative to the average index of weekly searches for the term in the country or state over the entire time period.

## 3. Empirical strategy

We estimate the impact of lockdown measures on family demography-relevant searches using both panel event-study and difference-in-differences methodology.

### 3.1 Panel event-study analyses

The panel event-study design is an econometric method for panel data that compares outcomes before and after an event: here the event is the beginning of the lockdown. It has been widely used in the economics literature for evaluating the impact of child birth on careers [53] and has been increasingly used by other disciplines to study the impact of various events in diverse domains (for example, air pollution [54]).

The policies implemented by different countries to slow the spread of COVID-19 were quite diverse; initially, the most common and strongest restrictions were stay-at-home orders. Abouk and Heydari [55] demonstrate that the initiation of "lockdown" restrictions had a strong causal impact on reduced social interactions. Because demographic behaviors are

strongly linked to social interactions, we define the beginning of the lockdown by the first stay-at-home order at the state (in U.S.) or country (in Europe) level.

A simple comparison of the number of searches before and after a lockdown would not account for seasonality in the number of searches for a given term, nor cross-country (U.S. state) differences in within-country (state) change in searches in the pre- and post-lockdown periods. As noted above, to account for seasonal variation and country (state)-level differences in weekly searches for a given term, we compare the relative index of search activity in a given week in a given country (state) in 2020 to the relative index of search activity for the same term in the same week in the same country (state) over the years 2016 to 2019. Seasonality is accounted for by including a full set of fixed effects for the week in the year $\gamma_w$, taking values 1 to 52. We also included in our models a full set of country-year fixed effects, which account for country-specific yearly changes in searches for a term. For instance, this would account for cross-country variation in the adoption of dating apps and websites.

We estimate the following model:

$$\tilde{R}_{cwyt} = \alpha + \delta_{yc} + \gamma_w + \sum_{\tau=-5}^{13} \beta_\tau 1\{t = \tau\} + \phi' x_{cwy} + \varepsilon_{cwyt} \tag{1}$$

where the outcome variable, $\tilde{R}_{cwyt}$ is the relative number of Google searches in country (state) $c$, during week $w$ of year $y$; $t$ counts the number of weeks before or after a lockdown and is equal to 0 the week the lockdown started; $\delta_{yc}$ is a full set of country (state)-year fixed effects; and $\gamma_w$ are week fixed effects. These country-year fixed effects capture most of the heterogeneity in google searches at the country*year level; however, we also performed robustness checks in which we introduced additional control variables adjusting for the severity of the epidemic in the country (state) at a given time.

As noted above, $t$ measures the number of weeks before or after a lockdown. We include binary variables indicating the week relative to the lockdown spanning 5 weeks before lockdown to 13 weeks after lockdown. The coefficients of interest are $\beta_\tau$ and are interpreted as the relative Google searches for a specific word in the given week before or after the lockdown, as compared to the same week of the year in the previous years of observation. They are interpreted as the average difference in relative searches in that week for the countries (states) included in the sample.

A set of binary variables $x_{cwy}$ adjust for country-specific events that may be related to a surge in the relative search interest for a given search term. Not controlling for such events would bias the estimation of seasonality. Doing so reduces "noise" in specific weeks that could result in measurement error bias. These events largely reflect searches due to demographic events experienced by celebrities (see discussion of outliers below).

We conducted our analyses separately for the European countries and U.S. states. Standard errors are clustered at the country (state) level. We therefore leverage two sources of variation: Google searches in the same week of previous years and the week in which lockdown started in each country or state.

## 3.2 Difference-in-differences analyses

In a second set of analyses, we estimate the impact of the lockdown using a difference-in-differences approach. We estimate a similar model to model (1), except that we now include a single binary variable which is coded "1" for the entire period after a lockdown is initiated (including any period in which the lockdown was no longer in effect), denoted as $1\{t \geq 0\}$, rather than the indicators for weeks preceding of following a lockdown. We estimate the

following model:

$$\tilde{R}_{cwyt} = \alpha + \delta_{yc} + \gamma_w + \beta * 1\{t \geq 0\} + \phi' x_{cwy} + \varepsilon_{cwyt} \tag{2}$$

Our coefficient of interest is $\beta$. It is interpreted as the average change in relative searches before and after the lockdown. While $\beta_\tau$ of the event study specification identifies the dynamic effect of the impact of the lockdown over time, $\beta$ gives the average impact in the post- relative to pre-lockdown period. Again, we estimate the model separately for the European countries and U.S. states. Standard errors are clustered at the country (state) level. This approach leverages three sources of variation: Google searches in the same week of previous years, the week in which lockdown started in each country (state), and, for the U.S., whether a lockdown was implemented in the state, given that not all states imposed mandatory lockdowns.

### 3.3 Outliers

As noted above, we also introduce a set of binary variables $x_{cwy}$ for particular weeks in a given year. This allows us to adjust for events that generated a large number of searches for a particular term. For instance, binary variables for week 19 and 20 of 2018 allow us to account for the tremendous worldwide interest in the Duke and the Duchess of Sussex's wedding, which resulted in a large number of Google searches for the search term "wedding".

We conducted three tests to empirically identify different types of outliers for each time series: (i) Chauvenet's Criterion [56], (ii) Hampel Filter [57], and (iii) outliers according to a Seasonal Trend decomposition using Loess [58]. Each test was conducted using the observations between 2016 and 2019 (inclusive), thus avoiding any influence of the potential effects of the lockdown, which we want to identify. S3 Table presents these results. We identify several events that received a tremendous interest worldwide (e.g., the Duke and Duchess of Sussex's wedding, as mentioned above, as well as changes in abortion law in the U.S., which generated a considerable increase in abortion-related searchers in European countries as well). Other events generated considerable local interest, such as weddings of celebrities.

This pattern highlights the importance of looking for outliers when using Google Trends data for the analysis of demographic topics. Failing to account for such outliers would introduce bias in our estimations, especially for the event study specification. Consider the Duke and the Duchess of Sussex's wedding: the event is associated with a large number of searches in week 19 and 20 of 2018, thus increasing the index of relative searches for the word "wedding". The week fixed-effect estimate, were we not to adjust for this event, would be too large as it would measure the seasonally-adjusted index of relative searches for the word "wedding" including the deviation due to the royal wedding. Week 19 and 20 in 2020 correspond to the 8[th] and 9[th] week after the beginning of lockdown in most European countries (7[th] and 8[th] week in the U.S.). If we do not adjust for this outlier, the coefficient for these weeks would be biased by the event. For a graphical depiction, see S1 Fig.

### 3.4 Preferred model and robustness checks

Our preferred model is the event-study model because it allows for observing variation in the timing of trends in demographically-relevant searches in each week before and after the lockdown. This model allows us to explicitly examine exacerbation and/or fade out of particular searches over time. In contrast, the difference-in-differences approach estimates only the average difference between the pre- and post-lockdown periods. The primary advantage of also estimating the difference-in-differences models is that, unlike the event study models, this specification allows us to include U.S. states that did not implement a lockdown, thereby providing state-specific effects of lockdowns relative to time trends in states that did not implement them.

We implemented several robustness checks. First, we estimated supplemental models in which we controlled for either the number of confirmed COVID cases or the number of COVID-related deaths in the state/country in a given week to account for the possibility that the severity of the epidemic at the country- or state-level may be associated with individuals' emotional states and thus influence their internet search behaviors. We selected these particular indicators because they are commonly reported by the media and subject to public discourse. Thus, trends therein may affect stress and uncertainty and, thereby, demographically-relevant behaviors. Controlling for either the number of COVID cases or COVID-related death slightly reduces the magnitude and (in rare cases) statistical significance of our estimates, but does not substantially change the pattern of effects.

Second, whereas in our preferred model specifications, we include an interaction of the country and year fixed effects, we also tested a specification in which year effects are not country specific. That is, we included separate year and country fixed effect rather than their interaction. Our results were robust to this specification. As a third robustness check, we used a log transformation of the raw index of relative searches provided by Google Trends in place of our primary outcome, for which we transformed the raw series to represent a deviation from the mean. Our results were robust to this specification, as well.

Fourth, to test whether our results are driven by search patterns in specific reference years, we changed the reference group of years following a "Leave-One-Out" cross-validation technique, dropping one or several years among the reference years 2016–2019 and re-estimating the models. Despite some slight variation in the estimates, the overall pattern of results remained consistent. It is reassuring that changing the base year does not systematically affect our results. Additionally, we performed a placebo test in which we excluded the year 2020 (in which the pandemic and associated lockdowns ensued) and considered placebo-lockdowns to have occurred on the same dates as the actual 2020 lockdown, but in 2019. We observe no impact of placebo-lockdowns on searches for most of the search terms, with the exception of searches related to abortion and the morning after/emergency pill in the U.S. However, this reflects that the abortion-related debate associated with the "Human Life Protection Act" primarily occurred in May 2019. Thus, our results indicate that the placebo-lockdowns were *positively* associated with search activity for abortion- and morning after/emergency pill-related searches in 2019 (an association we believe to be spurious). In contrast, the actual lockdowns had a *negative* impact on search activity for these terms. The difference in patterns in the intensity of search activity in this domain between Spring 2019 and Spring 2020 was likely driven by different underlying factors. All results from the robustness checks are available upon request.

Fifth, we tested the robustness of our results to the inclusion/exclusion of additional keywords associated with each search term. A first robustness check, for "condom" and "wedding" searches, for example consisted of adding the relevant English term in searches in non-English-speaking countries given that English has become a global language that is potentially used for Google searches in non-English speaking countries. For instance, the term "wedding planner" is commonly used in many non-English-speaking countries. Adding the English terms to the searches only slightly changed the results for European countries. It affected the magnitude of the point estimates most in Germany and Austria and least in Italy, Spain, and France, but did not change the overall patter of results. A second robustness check tested the sensitivity to the number of words by restricting the list of search terms to only those most commonly used. Again, this slightly changed the magnitude of the estimates but did not affect the overall pattern of results. The country- or state-specific difference-in-difference results are somewhat more sensitive to the inclusion/exclusion of some terms than the results for Europe or U.S. as a whole. These robustness checks indicate that our analyses are slightly sensitive to the inclusion/exclusion of keywords when: (i) several words can be used to refer to the same construct or (ii) the analysis relies on a small number of geographic areas.

Finally, we tested whether our results are driven by the (perceived) economic uncertainty resulting from the pandemic and the associated lockdowns. Specifically, we examined trends in Google searches for the term "unemployment" following initiation of a lockdown. We found a strong increase in the U.S., but heterogeneous patterns in European countries, suggesting stronger economic uncertainty in the U.S. As such, we estimated supplemental models in which we included the search intensity for the "unemployment" as a control. In Europe, the inclusion of this control had a negligible impact on the estimates. In the U.S., it had as somewhat larger influence on the magnitude of our estimates, such that they tended to be smaller when searches for "unemployment" was controlled. We will mention results impacted by the inclusion of the "unemployment" control below.

## 4. Results

### 4.1 "Lockdown" searches

We first tested our models using Google searches for the word "lockdown" to ensure that searches for lockdown increased around the date of lockdown implementation in each country or state. Our event study results, shown in Fig 1 (see also S4 Table for Europe and S5 Table for

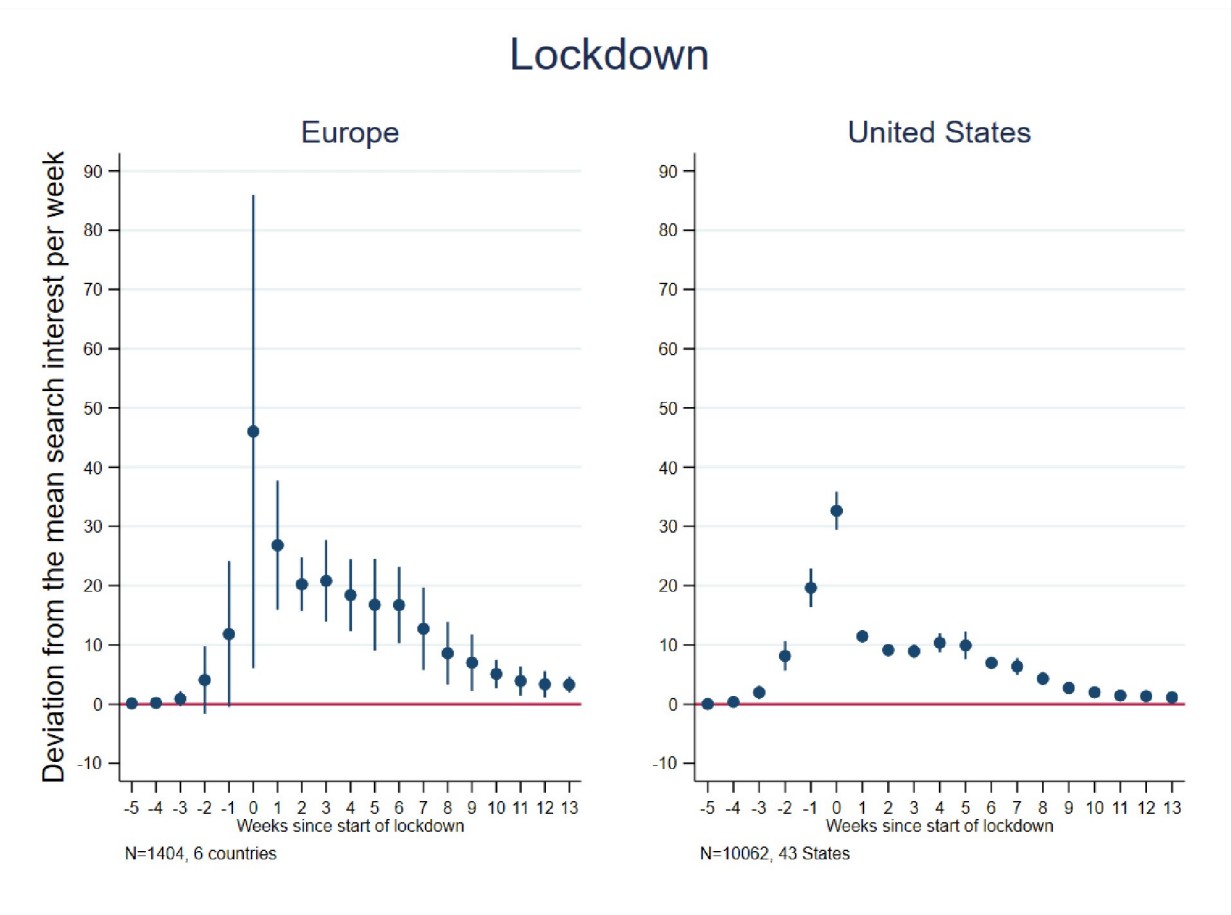

**Fig 1. Event study estimates of associations of lockdown timing with Google searches for lockdown.** Event study estimation results. The model for Europe included 6 countries: Austria, France, Germany, Italy, Spain, and the United Kingdom. The estimation sample for the United States included the 42 States that implemented a lockdown + Washington D.C. See S2 Table for lockdown dates and States included in the estimation sample.

**Table 1. Difference-in-differences estimates for lockdown-, family planning-, and fertility related search terms.**

|  | Lockdown | Condom | Emergency pill | Pregnancy test | Abortion | Plan Child | Plan other children |
|---|---|---|---|---|---|---|---|
|  | b/se | b/se | b/se | b/se | b/se | b/se | b/se |
| Europe | 12.68*** | -0.01 | -0.29*** | -0.04 | -0.06 | 0.20 | 0.01 |
|  | (1.12) | (0.07) | (0.03) | (0.02) | (0.02) | (0.18) | (0.03) |
| Observations | 1404 | 1404 | 1404 | 1404 | 1404 | 1404 | 1404 |
| US (English) | 5.25*** | -0.11*** | -0.19*** | -0.03* | -0.16*** | -0.00 | 0.08 |
|  | (0.45) | (0.02) | (0.05) | (0.01) | (0.02) | (0.05) | (0.05) |
| Observations | 11934 | 11934 | 10530 | 11934 | 11934 | 11466 | 11934 |
| US (Spanish) | 4.68** | -0.16 | NA | 0.06 | -0.06 | NA | 0.13 |
|  | (0.72) | (0.13) |  | (0.02) | (0.11) |  | (0.13) |
| Observations | 936 | 936 |  | 936 | 936 |  | 936 |
| Austria | 9.96*** | 0.22** | -0.30** | 0.02 | -0.02 | 0.12 | -0.03 |
|  | (2.22) | (0.07) | (0.09) | (0.07) | (0.08) | (0.09) | (0.19) |
| Observations | 234 | 234 | 234 | 234 | 234 | 234 | 234 |
| Germany | 15.04*** | 0.18*** | -0.18*** | -0.06 | -0.05 | 0.06 | -0.05 |
|  | (3.32) | (0.03) | (0.04) | (0.03) | (0.05) | (0.03) | (0.07) |
| Observations | 234 | 234 | 234 | 234 | 234 | 234 | 234 |
| France | 15.25*** | -0.16*** | -0.21*** | -0.03 | -0.03 | 0.18*** | -0.02 |
|  | (1.46) | (0.04) | (0.05) | (0.03) | (0.07) | (0.05) | (0.09) |
| Observations | 234 | 234 | 234 | 234 | 234 | 234 | 234 |
| Italy | 9.24*** | -0.06 | -0.32*** | -0.09* | -0.04 | -0.17 | 0.01 |
|  | (1.20) | (0.04) | (0.06) | (0.04) | (0.06) | (0.61) | (0.09) |
| Observations | 234 | 234 | 234 | 234 | 234 | 234 | 234 |
| Spain | 14.01*** | -0.16** | -0.34*** | -0.06 | -0.18* | 1.01 | 0.15 |
|  | (1.23) | (0.06) | (0.09) | (0.05) | (0.08) | (0.76) | (0.14) |
| Observations | 234 | 234 | 234 | 234 | 234 | 234 | 234 |
| U.-K. | 12.10*** | -0.10** | -0.35*** | -0.01 | -0.05 | -0.03 | 0.01 |
|  | (0.86) | (0.04) | (0.04) | (0.02) | (0.07) | (0.08) | (0.07) |
| Observations | 234 | 234 | 234 | 234 | 234 | 234 | 234 |

Note: Google Trends extraction made July 6, 2020. All models include controls for country-specific public events with implications for specific searches (see S3 Table). "NA" = "not available" because the number of searches in Google was not large enough to provide non null observations. There were 234 observation (weeks) for each of the European countries and each State in the U.S. Data for the U.S. include the 50 States and Washington, D.C., with the exception of models for "Plan children" for which Google only returned null observations for Wyoming and North Dakota and "Emergency pill" for which Google only returned null observations for Wyoming, Vermont, South Dakota, North Dakota, Montana, and Alaska. Spanish searches in the U.S. returned null searches in all states except California, Florida, New York, and Texas.

\* p < .05

\** p < .01

\*** p < .001.

the United States), confirm this: searches for "lockdown" began to increase in the 2–3 weeks preceding lockdown implementation, peaked in the first week of lockdown, and declined over time thereafter. The difference-in-differences estimates, shown in Table 1, also indicate that lockdowns resulted in statistically significant increases search interest for "lockdown" in Europe (and each sample country therein) and the U.S. In terms of effect sizes, the event study results indicate that the European countries averaged 46 times more relative searches for "lockdown" in the first week of their lockdown than in the comparable week in prior years, while the U.S. averaged 33 times more relative searches in the week of a lockdown. The differences-

in-differences estimates indicate that there was a weekly average of 13 times more searches for Europe and 5 times more searches for "lockdown" the U.S. in the period following lockdown.

Notably, the average magnitude of searches in the first week of lockdown was considerably larger in Europe than in the United States, likely reflecting that lockdowns tended to be more stringent in the European context. In addition, the standard errors are considerably larger in Europe than in the United States, reflecting substantial variation in searches across European countries. Larger standard errors for the European countries may also potentially reflect fewer estimation points in Europe (6 countries) compared to the U.S. (43–50 states), as well as variation in the day of the week (defined as spanning Sunday through Saturday) in which lockdown was initiated in a country (Notably, patterns of searches for "lockdown" across U.S. states were more similar in timing than across European countries. This may indicate that the U.S. population as a whole began searching for information about lockdowns when the first states began enacting them, whereas European searches tended to be more aligned with a lockdown being initiated in one's own country.). For example, Germany initiated lockdown at the beginning of the week and exhibited a substantial increase in searches for the term in that week, whereas Spain introduced lockdown at the end of the week and thus exhibited little increase in that week, but considerably greater increase in the subsequent week. On the whole, however, these findings strongly suggest that Google searches were used in both the European and U.S. contexts to obtain lockdown-related information around the time of lockdown initiation, thus supporting the validity of our empirical approach.

## 4.2 Sexual behavior-, contraceptive use-, pregnancy termination- and fertility-related searches

To consider the potential impact of pandemic-associated national (state) lockdowns on future fertility, as approximated by internet searches [59–61], we next examined searches relevant to sexual behaviors, contraceptive use, pregnancy termination and fertility, including those for (in U.S. English) condom, morning after/emergency pill, pregnancy test, abortion, plan child and plan other children (see S1 Table for the full list of search terms in each country). Figs 2–4 present event study results (their estimates are in S4 and S5 Tables for Europe and the U.S., respectively) and Table 1 presents difference-in-differences results for these searches.

The event study results for "condom" (Fig 2) indicate a relatively brief decline in Europe, centered around the first two weeks of lockdown, and a larger and more long-lasting decline in the United States. As such, the difference-in-differences estimates (Table 1) reveal no pre-post lockdown significant difference for Europe as a whole over the observation period, but an overall decline in relative searches for the U.S. for in searches in English, though not in Spanish. On average, there were 11 percent fewer searches for "condom" in the U.S. in the post-, relative to pre-lockdown period. It is also notable, however, that the difference-in-difference estimates reveal considerable heterogeneity within Europe, with decreases in France, Spain and the U.K. and an increase in searches in Germany and Austria where there had been rumours of a shortage of condoms during lockdown: Amid reports of a global shortage of condoms, with the world biggest producer forced to shut down production in Malaysia as a result of the pandemic, there were increased condom sales in Germany due to anticipated forthcoming restrictions [62].

In contrast to the results for "condom," we find a large decline in relative searches for emergency contraception (Fig 1) in the wake of lockdowns, particularly in Europe, with effects persisting for more than two months post-lockdown introduction and reaching a maximum magnitude of roughly 50 fewer searches. This may imply fewer incidents of unprotected sex or contraceptive failures, likely driven by a decrease in sexual activity among younger people and

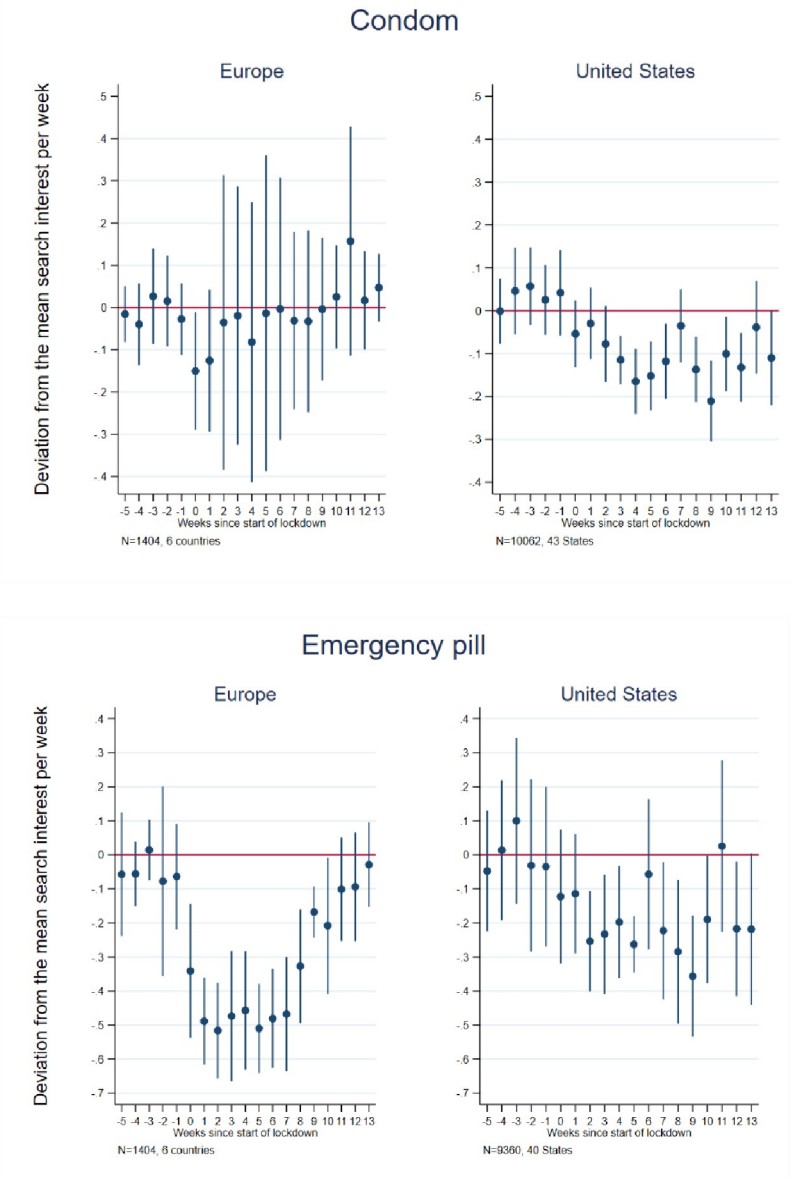

**Fig 2. Event study estimates of associations of lockdown timing with Google searches for condom and morning after/emergency pill.** Event study estimation results. The model for Europe included 6 countries: Austria, France, Germany, Italy, Spain, and the United Kingdom. The estimation sample for the United States included the 42 States that implemented a lockdown + Washington D.C. See S2 Table for lockdown dates and States included in the estimation sample.

those in non-cohabiting relationships. While the U.S. also demonstrates an overall decreasing trend in the relative searches for emergency contraception in the first three months after lockdowns, the drop was less severe and consistent across weeks; however, unlike in Europe, it showed little sign of return to average pre-lockdown search levels by 13 weeks post-lockdown. The difference-in-differences estimates (Table 1) indicate 29 percent fewer searches in Europe and 19 percent fewer searches in the U.S. (in English since Spanish-language searches in the U.S. did not reach Google Trends' minimum thresholds and could not be included in this analysis) in the post- than pre-lockdown periods. The decline in emergency contraceptive searches

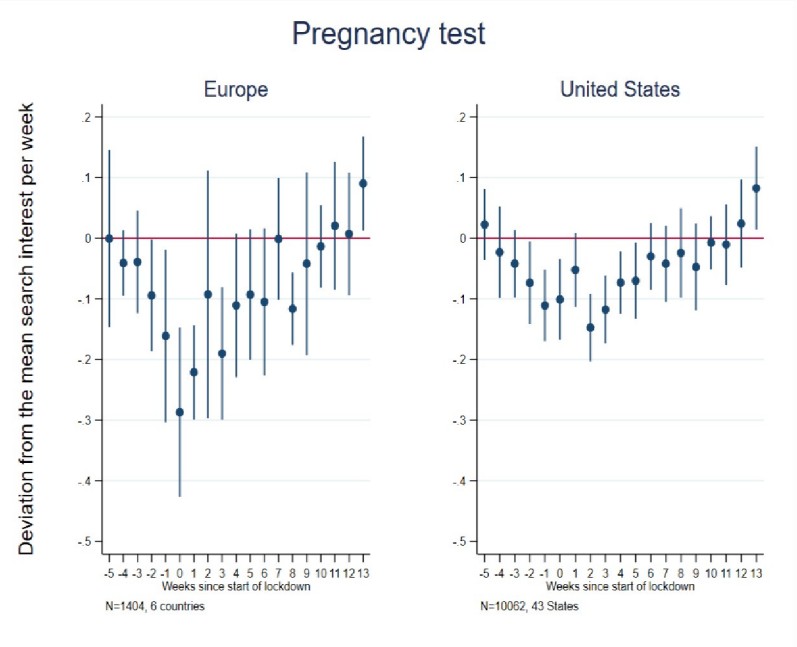

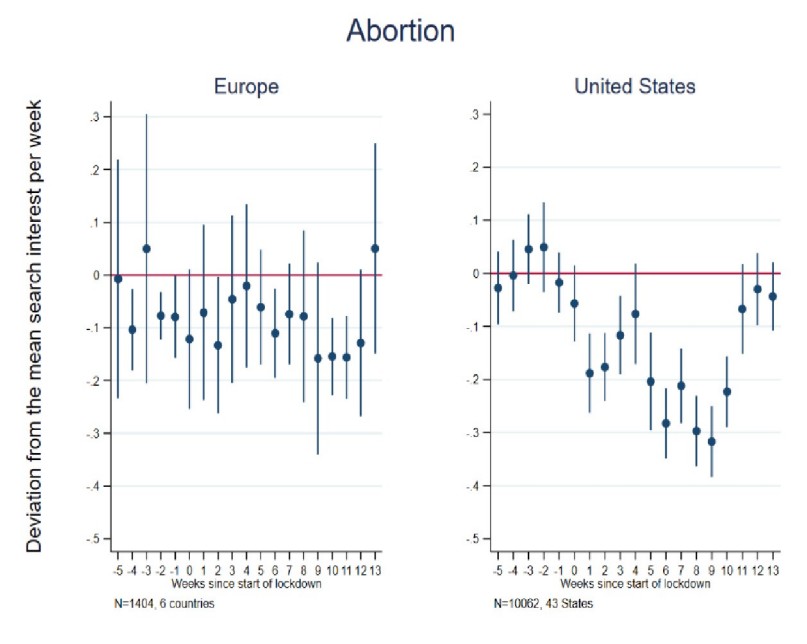

**Fig 3. Event study estimates of associations of lockdown timing with Google searches for pregnancy test and abortion.** Event study estimation results. The model for Europe included 6 countries: Austria, France, Germany, Italy, Spain, and the United Kingdom. The estimation sample for the United States included the 42 States that implemented a lockdown + Washington D.C. See S2 Table for lockdown dates and States included in the estimation sample.

was somewhat larger in magnitude in Austria, Italy, Spain, and the U.K. than in Germany, France, and the U.S.

Event study results for "pregnancy test" and "abortion" are presented in Fig 3 (See S4 and S5 Tables for point estimates and standard errors for Europe and the U.S., respectively).

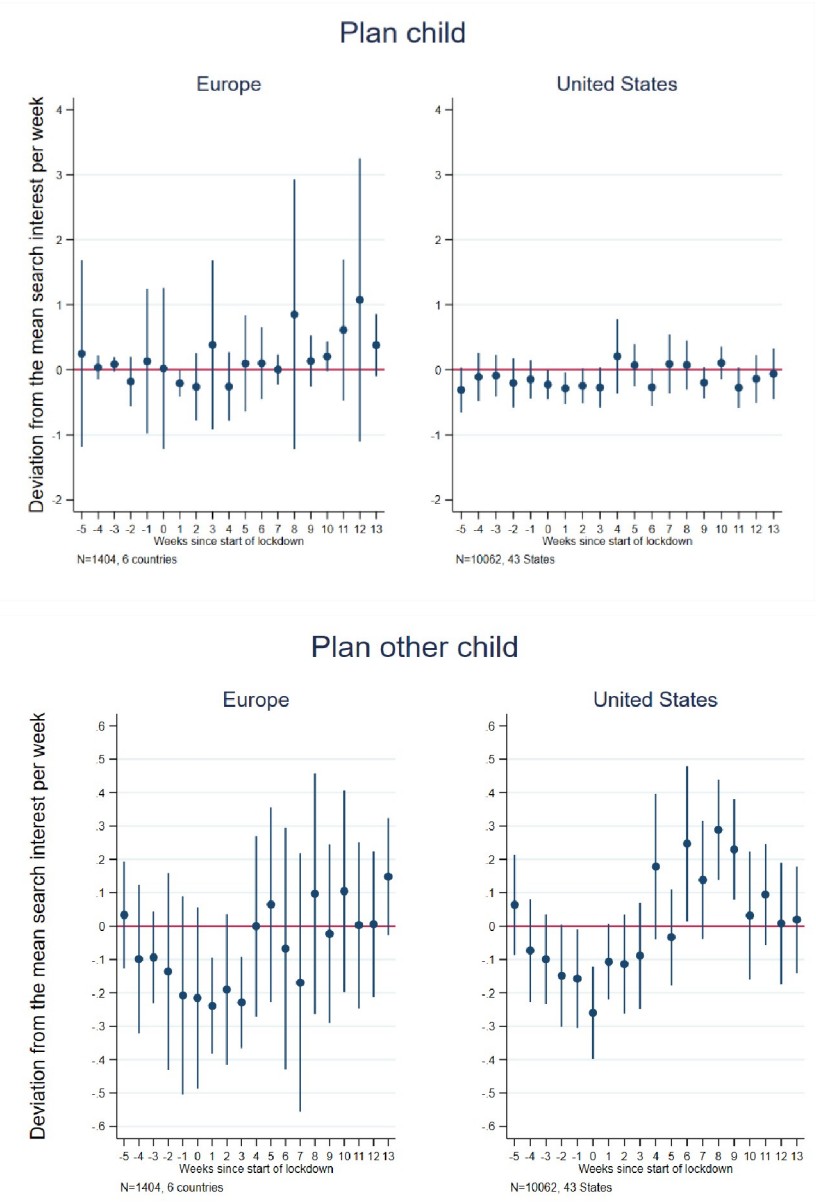

**Fig 4. Event study estimates of associations of lockdown timing with Google searches for planning for a birth/child.** Event study estimation results. The model for Europe included 6 countries: Austria, France, Germany, Italy, Spain, and the United Kingdom. The estimation sample for the United States included the 42 States that implemented a lockdown + Washington D.C. See S2 Table for lockdown dates and States included in the estimation sample.

Relative searches for terms related to pregnancy tests follow the same general trend in the U.S. and Europe, though of more pronounced magnitude in the latter: a modest decrease in the early weeks of the lockdown period followed by a return to average search levels thereafter and, ultimately, a slightly higher than usual level by the 13th week after a lockdown. The difference-in-differences results (Table 1) suggest no overall statistically significant pre-post lockdown difference across the observation period for Europe as a whole (with a significant decline of 9 percent found only for Italy), and a small overall post-lockdown decline in U.S. searchers (in English) of 3 percent.

Turning to abortion-related searches, in the event study, we observe some variation, but no clear pattern of substantial decline in Europe, with the potential exception of a significant dip roughly 10–12 weeks after lockdown initiation. In contrast, we observe a substantial and statistically significant decline (though with some variation therein) in the U.S. beginning at the initiation of lockdown and lasting approximately 10 weeks (Fig 3). Thus, the difference-in-differences results indicate an overall decline in abortion-related search intensity (in English) in the U.S. of 16 percent in the post-, relative to pre-, lockdown period, but no difference for Europe as a whole. With respect to the individual European countries, only Spain exhibits a significant change, with an 18 percent decline in abortion-related search intensity. Again, this may reflect a decrease in (unprotected) sexuality activity, perhaps particularly among those not experiencing lockdown with a coresident partner [63].

With respect to birth planning, the event study (Fig 4) results provide little evidence of an impact of lockdown on planning for any child in Europe or the U.S. Likewise, the difference-in-differences estimates (Table 1) are nonsignificant for both. This result is in line with findings from [9], which showed that fertility intentions during the lockdown decreased, although less so in France than Spain and Italy, mainly reflecting a postponement of fertility plans rather abandonment thereof. Notably, however, we do observe a significant 18 percent increase in such searches in France in the post-lockdown period. This may mean that lockdown will have positive effects on fertility in France, which has particularly generous family policies. For searches specifically focused on planning for higher order births ("other child," "second child," "third child") we observe in the event study a slight and short-lived decline in relative searches in Europe in the weeks immediately surrounding the lockdowns, and some hint of a modest and short-lived increase in searches in the U.S. beginning 4–6 weeks after lockdowns, but returning to average by week 10 post-lockdown. The difference-in-differences estimates reveal no significant finding for Europe (or any sample country therein) or the U.S., except when we control for searches for "unemployment," in which case lockdown is associated with increased searches related to additional children and to divorce.

## 4.3 Relationship- and family formation-related searches

In order to explore the possible impact of the lockdowns on potential relationship- and family formation behaviors, we consider trends in searches for terms relevant to dating, relationships, and marriage. Searches relating to dating applications and dating websites trended toward declines in Europe in the weeks preceding and following lockdown initiations, though rarely attained statistical significance in any week in the event study (Fig 5 and estimates in S6 and S7 Tables for Europe and the U.S., respectively). Because many users of dating applications use them directly rather than through internet searches, our analysis may substantially undercount actual usage of such applications. Also, it may be most relevant to potential or "new" users who are searching for applications to which to subscribe rather than to current subscribers of such applications Nor did we observe a significant effect in the difference-in-differences analysis (Table 2) for the sample European countries as a whole, though Italy and Spain demonstrated a decline in relative searches of 60 and 16 percent, respectively, in the post-lockdown period. The declining trend in relative searches in the event study, while smaller in magnitude, tended to be statistically significant in the U.S. in the weeks preceding and following the lockdown. This pattern is affirmed by the difference-in-differences results which indicated a 6 percent decline in English searches and 9 percent decline in Spanish searches in the post-lockdown period. Also, unlike in Europe, no return to pre-lockdown search levels is observed in the U.S., perhaps because COVID infection levels continued to accelerate in the U.S. compared to Europe over the observation period. A decline in the use of dating applications has been observed in Australia in the wake of the pandemic [63].

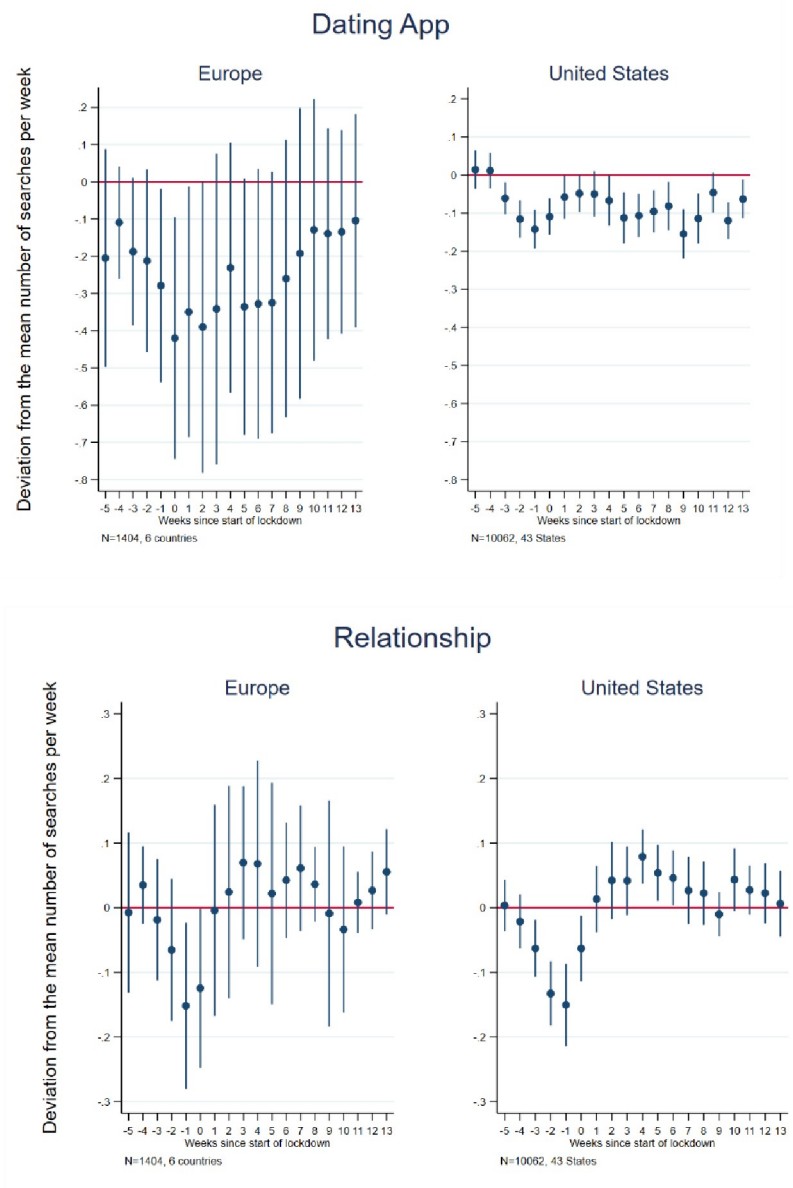

**Fig 5. Event study estimates of associations of lockdown timing with Google searches for dating apps/websites and relationship/couple.** Event study estimation results. The model for Europe included 6 countries: Austria, France, Germany, Italy, Spain, and the United Kingdom. The estimation sample for the United States included the 42 States that implemented a lockdown + Washington D.C. See S2 Table for lockdown dates and States included in the estimation sample.

Searches for words related to coupling and romantic relationships showed a slight and short-lived decline in Europe in the weeks preceding the lockdown (Fig 5); the difference-in-differences analyses (Table 2) failed to produce a significant overall results, although France and Italy exhibited an increase of 10 and 13 percent, respectively, in such searches in the post-lockdown period. The U.S. exhibited a decline is such relative searches beginning about 3 weeks before lockdowns and lasting roughly until the first week after their initiations, before returning to pre-lockdown or slightly above pre-lockdown levels. Thus, the difference-in-differences results suggest a 6 percent increases in such (English) searches in the post-lockdown period.

**Table 2. Difference-in-differences estimates for lockdown-, family planning-, and fertility related search terms.**

| | Wedding | Dating App. | Relationship | Divorce | Break up |
|---|---|---|---|---|---|
| | b/se | b/se | b/se | b/se | b/se |
| Europe | -0.41*** | -0.15 | 0.04 | -0.11* | 0.03 |
| | (0.04) | (0.09) | (0.03) | (0.03) | (0.07) |
| Observations | 1404 | 1404 | 1404 | 1404 | 1404 |
| US (English) | -0.14*** | -0.06*** | 0.06*** | -0.01 | 0.13*** |
| | (0.01) | (0.02) | (0.01) | (0.01) | (0.02) |
| Observations | 11934 | 11934 | 11934 | 11934 | 11934 |
| US (Spanish) | -0.30*** | -0.09* | 0.03 | -0.00 | 0.17 |
| | (0.02) | (0.02) | (0.03) | (0.06) | (0.12) |
| Observations | 936 | 936 | 936 | 936 | 936 |
| Austria | -0.44*** | -0.11 | 0.01 | -0.07 | -0.02 |
| | (0.06) | (0.06) | (0.05) | (0.07) | (0.10) |
| Observations | 234 | 234 | 234 | 234 | 234 |
| Germany | -0.43*** | -0.01 | 0.02 | -0.11* | -0.14 |
| | (0.07) | (0.04) | (0.02) | (0.05) | (0.07) |
| Observations | 234 | 234 | 234 | 234 | 234 |
| France | -0.37*** | -0.03 | 0.10** | -0.12** | 0.17* |
| | (0.03) | (0.03) | (0.03) | (0.04) | (0.08) |
| Observations | 234 | 234 | 234 | 234 | 234 |
| Italy | -0.46*** | -0.60*** | -0.04 | -0.20* | -0.12 |
| | (0.07) | (0.07) | (0.03) | (0.09) | (0.07) |
| Observations | 234 | 234 | 234 | 234 | 234 |
| Spain | -0.54*** | -0.16*** | 0.13*** | -0.14 | -0.05 |
| | (0.11) | (0.03) | (0.04) | (0.08) | (0.08) |
| Observations | 234 | 234 | 234 | 234 | 234 |
| U.-K. | -0.23*** | -0.02 | 0.04 | -0.02 | 0.29*** |
| | (0.04) | (0.03) | (0.02) | (0.04) | (0.05) |
| Observations | 234 | 234 | 234 | 234 | 234 |

Note: Google Trends extraction made July 6, 2020. All models include controls for country-specific public events with implications for specific searches (see S3 Table). "NA" = "not available" because the number of searches in Google was not large enough to provide non null observations. There were 234 observation (weeks) for each of the European countries and each State in the U.S.Data for the U.S. include the 50 States and Washington, D.C. Spanish searches in the U.S. provided returned null searches in all states except California, Florida, New York, and Texas.

* p < .05

** p < .01

*** p < .001.

Fig 6 shows event study results related to marriage and weddings. We see a steep decline in Europe beginning a few weeks before lockdowns and continuing for about 10 weeks post-lockdown initiations, before starting to level off. The difference-in-differences estimate for Europe as a whole indicates a 41 percent decline in relative searches in the post-lockdown period. For the individual countries, the magnitude of decline ranged from 54 percent (Spain) to 23 percent (U.K.). The U.S. exhibited a less steep decline, reaching a low of 28 percent below average in the week of lockdown before slightly ticking upward and levelling off well below average for the remainder of the period. The difference-in-differences estimates indicate a 14 percent pre-post decline in English searches and 30 percent pre-post decline in Spanish searches.

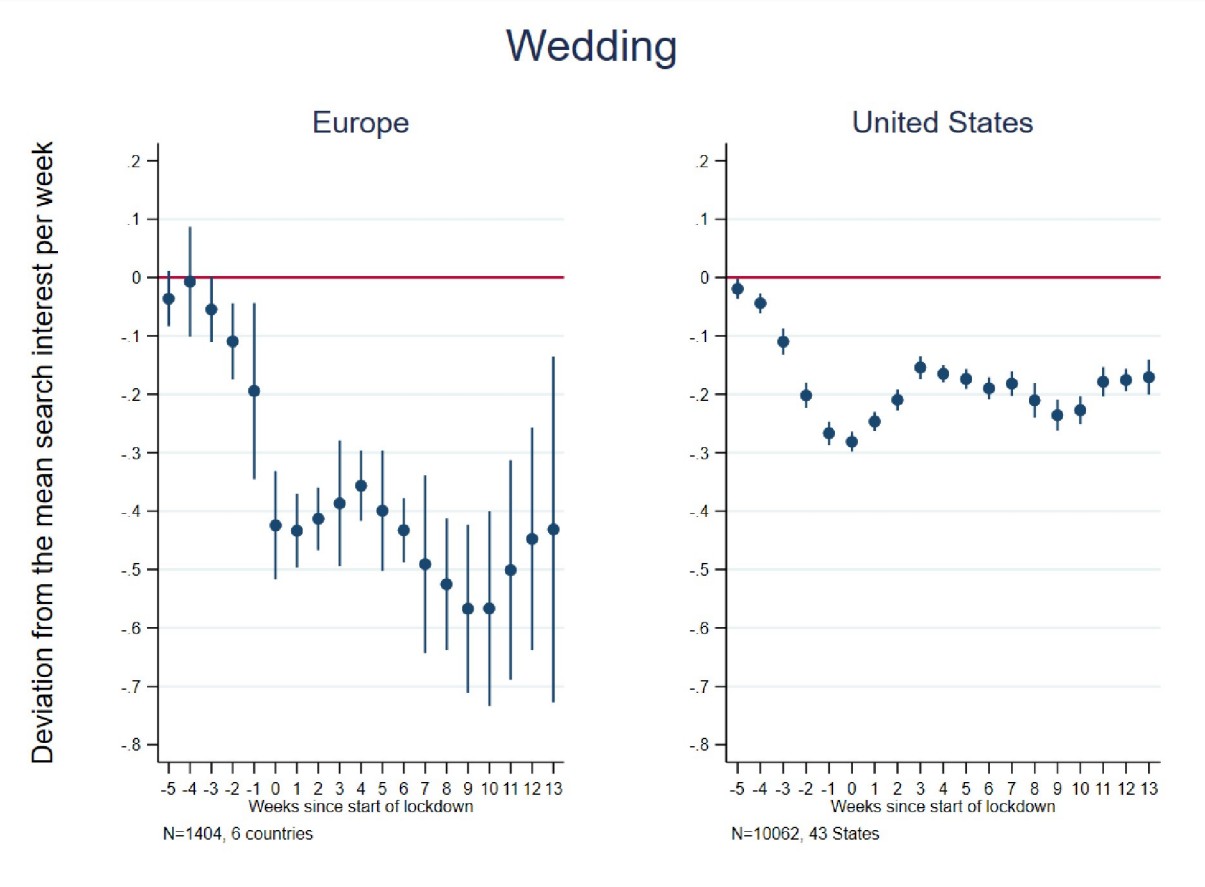

**Fig 6. Event study estimates of associations of lockdown timing with Google searches for marriage/wedding.** Event study estimation results. The model for Europe included 6 countries: Austria, France, Germany, Italy, Spain, and the United Kingdom. The estimation sample for the United States included the 42 States that implemented a lockdown + Washington D.C. See S2 Table for lockdown dates and States included in the estimation sample.

### 4.4 Union-dissolution related searches

Our event study results (Fig 7 and S6 and S7 Tables) show that relative searches for terms linked to romantic relationships "break-up" declined modestly in Europe in the weeks immediately adjacent to implementation of lockdowns before quickly returning to average pre-lockdown levels. We found no significant overall difference-in-differences estimate (Table 2) for the European countries, though France exhibited a 17 percent increase and the U.K. a 19 percent increase in break-up-related searches in the post-lockdown period. In contrast, the event study results for the U.S. indicate no change in break-up-related searches prior to lockdowns, but thereafter generally remained elevated (with some variation) throughout the subsequent period. The difference-in-differences estimate for the U.S. suggests a 13 percent increase in such relative searches in the post-lockdown period.

Finally, the event study results reveal declines in divorce-related relative searches in both Europe and the U.S., with a notably steeper decline in Europe than the U.S., beginning in the month preceding lockdowns and persisting for 1–2 months thereafter before returning to pre-pandemic average levels. We also observe a slight increase in divorce related searches in the U.S., beginning roughly 10 weeks after lockdown initiations (Fig 7). The difference-in-differences estimate for Europe indicates an 11 percent decline in relative searches in the post-lockdown

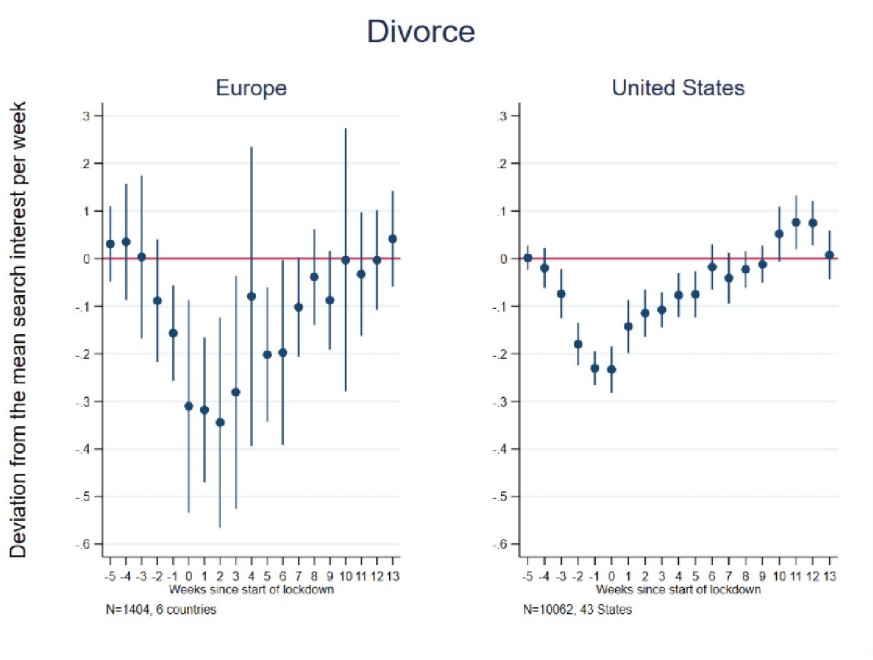

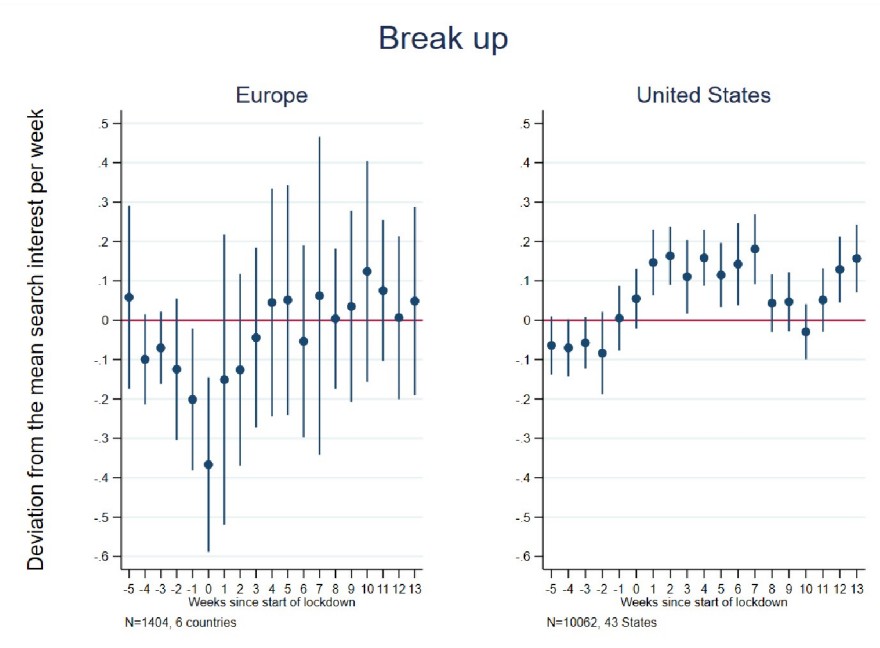

**Fig 7. Event study estimates of associations of lockdown timing with Google searches for breakup and divorce.**
Event study estimation results. The model for Europe included 6 countries: Austria, France, Germany, Italy, Spain, and the United Kingdom. The estimation sample for the United States included the 42 States that implemented a lockdown + Washington D.C. See S2 Table for lockdown dates and States included in the estimation sample.

period, with Germany (11 percent), France (12 percent), and Italy (20 percent) individually demonstrating significant relative declines. The difference-in-differences estimate for the U.S. is non-significant, likely reflecting a relative decline in such searches in the early part of the

post-lockdown period, followed by a subsequent increase. Interestingly, however, when searches for "unemployment" are controlled, we find that lockdowns in U.S. states had a positive impact on divorce-related searches in the difference-in-differences analyses. This may suggest that, in the absence of economic uncertainty, divorce searches would have been much higher. This is in line with work showing that divorce declines during economic crisis or downturn in the U.S. [64, 65].

## 5. Conclusion

The COVID-19 pandemic is likely to have a variety of short-, medium-, and long-term consequences for society that may vary across nations, at least in part, in accordance with governmental responses *vis-à-vis* lockdowns and other health and social policies. Notably, we hypothesized that consequences for demographic behaviors might be more marked in countries with less well managed outbreaks and with fewer safety nets, exacerbating feelings of uncertainty and economic insecurity. Pandemic-related research to date has primarily focused on mortality [2], migratory patterns [3, 4], economic wellbeing and government responses thereto [1, 15, 38, 45–48, 66], and physical and mental health and wellbeing [11–13, 36–38, 46, 48, 63]. [1] provide a comprehensive review of the pandemic-related research to date. Understanding more fully how population dynamics and, in particular, those related to family demography, may be affected is crucial to predicting subsequent demographic trends. To this end, we use Google Trends data and event study and difference-in-differences techniques to examine the influence of national and, in the case of the U.S., state-level lockdowns during the pandemic on demographically-relevant Google search trends. We focus specifically on sexual behavior-, contraceptive use-, pregnancy termination- and fertility-related searches; coupling, romantic relationship, and union formation-related searches; and union dissolution-related searches. On the whole, we find modest evidence of changes in search patterns related to family planning, but little evidence of changes in search patterns related to fertility, with the exception that current parents may have increased searches for information about having subsequent children. Using Google Trends data, Wilde et al. [61] predict a 15% decrease in fertility; these results are based on keywords linked to unemployment rather than fertility, with the assumption of a causal link between unemployment and fertility, as reported in previous years. When based solely on fertility key-related words, their results suggest no large negative effect on fertility and are in line with our results, in spite of using different fertility key-related. We do find some changes in search patterns related to relationship and union formation, as well as union dissolution. However, these tended to attain statistical significance only in the period immediately surrounding lockdown initiation and to return to average pre-pandemic levels within 2 to 3 months, particularly in the European context; divergence in U.S. trends tended to last longer. This fits with our initial hypothesis that lockdowns will lead to more marked demographic consequences in countries that managed the economic fallout from the pandemic less well and provided weaker financial safety nets.

Starting with our results for sexual behavior-, contraceptive use-, pregnancy termination- and fertility-related searches, we conclude that the introduction of lockdown measures resulted in a short-lived decline in relative searches for condoms in the U.S., but no change in Europe, and that searches for the morning after/emergency pill declined substantially in both Europe and the U.S. at the initiation of lockdowns, returning to average levels in Europe within a few months, but remaining below average after 3 months in the U.S. In both Europe and the U.S., relative searches related to "pregnancy test" declined slightly a few weeks prior to lockdowns but returned to average within 6 to 8 weeks of lockdown initiations. Searches for abortion declined in the U.S. and, to a much lesser extent in Europe, with substantial rebound

within 3 months of lockdown initiations. These findings may indicate some changes in sexual behaviors during lockdowns, such that unprotected sex and/or contraceptive failure may have temporarily declined, likely predominantly among non-coresident couples, during the early months of the pandemic and associated lockdowns, but tended to rebound relatively quickly.

We find no change in relative searches regarding overall planning for childbirth associated with lockdowns and their timing, but some suggestive evidence that couples who are already parents may take the opportunity of lockdown to consider having an additional child. We can place our results within the wider literature on drivers of fertility trends. First, these results speak to the literature on the impact unemployment or job insecurity [19, 22, 23]. Second, they fit with the literature of the response of fertility levels to "shocks", such as the Great Depression (which resulted in a "baby bust"), or World War II (and its resulting "baby boom"). Seminal work by Rindfuss, Morgan, and Swicegood [25] found, for instance, that economic recessions often lead to a postponement of childbearing, especially for first births. The literature emphasizes the importance of actual versus anticipated financial and employment situations. While actual financial losses and unemployment do correlate with fertility trends, an important pathway to understand the impact of macro-level business cycles on fertility is consumer confidence [20, 27]. Financial incertitude might be crucial to understanding fertility responses to the lockdowns: while unemployment levels have increased as a result of lockdowns, our results are often stronger in the very first weeks before or after lockdowns, that is, before any substantial impacts on jobs and income losses. Yet, we find no evidence that initial lockdowns lead to significant changes in overall fertility intentions, and only a slight suggestion that higher order births may be being considered by current parents. Our conclusions, however, only relate to fertility intentions (rather than realization) and initial lockdown implementation (rather than potential subsequent economic recessions or the length of the lockdown). Interestingly, the (small) potential impact of lockdowns on higher-order fertility suggested by our results show a relative decline in searches related to higher-order childbearing in the month preceding and the month following lockdowns, with the suggestive evidence of an increase in searches only appearing thereafter. It is possible that this pattern is, at least partially, driven by expectations of future financial and employment instability, particularly if such uncertainty was highest in the earliest months of the pandemic.

Turning to our results on relationships, union formation, and union dissolution, relative searches related to dating (and for dating/relationship applications) and coupling/relationship formation appear to be negatively affected by lockdowns. However, these impacts were often short-lived and, again, concentrated in the weeks immediately surrounding the introduction of shelter-in-place measures. However, fitting in with our initial hypothesis, we find some differences between the U.S. and Europe in these impacts and, in particular, their timing. In Europe, relative searches returned to pre-lockdown levels after only a modest change, whereas changes in searches in the U.S. appeared to be more long lasting. As we had hypothesized, this may reflect that, around the time of national lockdowns, the pandemic was considered to be better controlled in many European settings that in the U.S., and also that economic consequences may have been expected be better attenuated by stronger welfare state policies and faster and more comprehensive economic responses in Europe.

European and U.S. trends were similar for searches related to union formation, with relative declines in searches related to marriage and weddings that persisted in both contexts even three months after lockdowns were initiated, though the magnitude of decline was larger in Europe. To the extent that trends in union formation parlay into subsequent trends in fertility, this may suggest slight declines in near-term first-birth fertility. However, the direction of causality between marriage and fertility is ambiguous. That is, lower marriage rates may reflect individuals or couples not (yet) wanting a child and therefore choosing to not (yet) marry or

enter into cohabitation, rather than fertility being driven by changes in marriage or cohabitation formation behaviors [28, 29]. Moreover, delaying or opting out of marriage will only have an effect on fertility in countries where the tie between marriage and childbearing is strongest [28]. Furthermore, longer-term fertility and family formation plans will also be affected by post-lockdown economic and policy landscapes, which we do not observe.

It is less clear how lockdowns and resulting economic downturns might affect separation and divorce. Two contrasting hypotheses can be made [31]: economic instability increases financial and psychological stress for couples, which increase risk of separation and divorce, and, given considerable financial costs of separation and divorce (including legal costs and the loss of economies of scale), couples might be "priced out" of being able to separate during financially unstable or uncertain times. While evidence suggests that well-being might have decreased and stress increased during the pandemic [36], our results point—if anything— toward near-term declines in divorce, which is consistent with recent evidence from the 2008 recession, which resulted in couples putting off divorcing [30, 32]. Indeed, we find a modest short-lived decline in divorce-related relative search interest in Europe and a small short-lived decline in the U.S., particularly in the first weeks of lockdowns. Notably, however, we find little evidence of a change in searches for break-up in Europe and a slight, short-lived increase in break-up-related search interest in the U.S. in the wake of the lockdowns.

Overall, differences in demographically-relevant searches in the U.S. and Europe may reflect both the differential handling of the pandemic in the spring of 2020 and differences in underlying social welfare safety nets. Indeed, international heterogeneity in individual perceptions of the effects of the pandemic is remarkable—and tends to vary with both of these factors. For example, while threat perception of Covid-19 to oneself, one's family, one's local community, one's country, and the world were high across most countries at the start of the pandemic, they decreased over time in countries such as Germany and Italy, while increasing in the U.K. and U.S. [39]. Similarly, confidence in institutions such as the health system was lowest in the U.K. and U.S., and highest in European countries, particularly Spain [39]. Also, the economic consequences of the pandemic and the resulting lockdowns have not been shared equally within societies [66]. Of particular concern, inequality in job losses during the pandemic have been found to be particularly stark in the U.S. and U.K. [66]. If a main pathway to understanding possible impacts of lockdowns on future family demographic trends is economic hardship and uncertainty, then we might expect increasingly polarized behaviors, particularly in the U. S. where our results, though still small-to-modest in magnitude, tend to be stronger and longer-lasting than in Europe. Trends in increasing polarization of family formation behaviors have been well underway in the U.S. for several decades [67]. Of course, only time will tell what, if any, influence the pandemic and resulting economic downturn with have on family demographic trends, and where.

Our results must be interpreted in the light of a number of limitations. First, although Google search trends cannot be assumed to fully represent the interests, concerns, or preoccupations of a nation's population as a whole [65, 68], the popularity of Google as a search engine in Europe and the U.S., combined with quick and free data access, makes it a useful tool for observing immediate changes in search behaviors in response to the pandemic. Zagheni and Weber [69] recommend using difference-in-differences techniques, such as those used in this study, to assess relative trends in internet search data over time as an efficient means of reducing bias when the population-representativeness of internet searches by a subset of the population is unknown.

Second, as discussed in detail above, Google Trends only reports the relative (not absolute) search trends, specifically the number of searches for a particular term at a given time in a particular geographic unit *relative to the total search volume* in the geographic unit at a given time. This search index is then normalized to the maximum search volume observed over the

period in the geographic unit. In a usual period, as the number of total Google searches is very large in comparison to the number of searches for a given term, a small change in the denominator (all searches) is negligible for the interpretation of the index. However, some periods, particularly those characterized by large shocks—like the introduction of national lockdowns —may generate a large increase in overall Google searches. In this case, the variation in the total number of Google searches might influence the relation of particular searches to the relative search index. This may be particularly true during lockdowns when individuals are required to stay at home and may therefore be more likely to use the internet. This implies that, while the raw number of searches for a particular term may be constant over a period, the number of searches relative to the overall number searches will change if the total number of Google searches increases (decreases) over the period. While we suspect that the total volume of Google searches increased during lockdown, at the time of writing Google could not confirm this hypothesis. Google Trends data must, therefore, be interpreted as reflecting searches for a specific term relative to all Google searches, and not as reflecting absolute search intensity, *per se*. Nonetheless, a change in relative search intensity during lockdowns can be identified and evaluated.

Third, Google searches do not directly measure behaviors. Future research based on actual levels of demographic behaviors is needed to corroborate whether and how such searches may predict trends in family demography.

Fourth, our results represent average national/state level searches and may mask considerable heterogeneity in the way countries (and states) imposed lockdown restrictions, which in turn may influence perceptions and reactions to these measures. For country comparison purposes, we use a binary treatment: whether the lockdown was implemented or not. This choice does not allow differentiating the extent of different lockdowns in terms of length or stringency, nor considering other forms of restrictions than stay-at-home orders or their changing nature over time. Our analytical framework also does not allow considering intra-country variation within the European countries considered. Further decompositions by smaller geographical areas are warranted.

In sum, using Google Trends data, we find evidence that pandemic-induced lockdowns appear to have small and likely short-lived impacts on family demography-related searches, particularly in the U.S. We interpret these findings within the context of important economic and social policy variation across countries, such that social and economic uncertainty was likely of larger magnitude in the U.S. than Europe. Whereas it is possible that such uncertainty may result in potentially (small-to-modest in magnitude) impacts of the pandemic on future family demography in the U.S., whether any such impacts occur is a question for future research.

## Supporting information

**S1 Table. Search terms by topic, country, and language.**
(DOCX)

**S2 Table. Lockdown dates in European countries and U.S. states.**
(DOCX)

**S3 Table. Outlier search results excluded based on public interest events driving searches.**
(DOCX)

**S4 Table. Event study estimates for lockdown-, family planning-, and fertility related search terms, European countries.**
(DOCX)

**S5 Table. Event study estimates for lockdown-, family planning-, and fertility related search terms, United States.**
(DOCX)

**S6 Table Event study estimates for union formation- and union dissolution-related terms, European countries.**
(DOCX)

**S7 Table. Event study estimates for union formation- and union dissolution-related terms, United States.**
(DOCX)

**S1 Fig.** Deviation from the mean Google search interest for wedding and marriage related terms by U.S. state (top panel) and European country (bottom panel).
(DOCX)

# Acknowledgments

Note: We are grateful to Mathis Sansu of the École Nationale de la Statistique et de l'Administration Économique (Ensae) for excellent research assistance and to the following colleagues for providing in-country relevant translations from U.S. English into their native tongues: Eva Beaujouan and Caroline Berghammer for German in Austria, Marie Bergstrom and Caroline Uggla for Swedish in Sweden, Suzanne Eckhardt and Felix Tropf for German in Germany, and Alejandra Ros-Pilarz for (Mexican) Spanish in the United States. French, Italian, Spanish (Spain), and UK-English were translated by our team. We are also thankful to Emanuele Del Fava for providing support with the R script and Google Trends API.

# Author Contributions

**Conceptualization:** Lawrence M. Berger, Giulia Ferrari, Marion Leturcq, Lidia Panico, Anne Solaz.

**Writing – original draft:** Lawrence M. Berger, Giulia Ferrari, Marion Leturcq, Lidia Panico, Anne Solaz.

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
