## [Decision Letter · Decision Letter 0]

23 Dec 2020

PONE-D-20-38713

COVID-19 Lockdowns and Demographically-Relevant Google Trends: A Cross-National Analysis

PLOS ONE

Dear Dr. Solaz,

Thank you for submitting your manuscript to PLOS ONE. After careful consideration, we feel that it has merit but does not fully meet PLOS ONE’s publication criteria as it currently stands. Therefore, we invite you to submit a revised version of the manuscript that addresses the points raised during the review process.

The paper is perceived by the three reviewers as well written and of high technical quality but there are some clarifications needed on the setup, the details of the analysis and interpretation. In addition, PLOS ONE uses the Vancouver system of citations and the manuscript will have to be adapted to it before being accepted for publication.

We look forward to receiving your revised manuscript.

Kind regards,

José Antonio Ortega, Ph.D.

Academic Editor

PLOS ONE

Journal Requirements:

Reviewers' comments:

Reviewer's Responses to Questions

**Comments to the Author**

1. Is the manuscript technically sound, and do the data support the conclusions?

Reviewer #1: Yes

Reviewer #2: Partly

Reviewer #3: Partly

2. Has the statistical analysis been performed appropriately and rigorously? 

Reviewer #1: Yes

Reviewer #2: I Don't Know

Reviewer #3: Yes

3. Have the authors made all data underlying the findings in their manuscript fully available?

Reviewer #1: Yes

Reviewer #2: No

Reviewer #3: Yes

4. Is the manuscript presented in an intelligible fashion and written in standard English?

Reviewer #1: Yes

Reviewer #2: Yes

Reviewer #3: Yes

5. Review Comments to the Author

Reviewer #1: This is a well-written and well-illustrated paper. Draws on big data sources from Google Trends, the paper uses event study and difference-in-differences statistical techniques and examined trends of people’s search interests related to demographically-relevant terms during pre- and post- COVID19 lockdowns in Europe and U.S. states. The findings have important managerial implications for decisionmakers and researchers.

It is suggested that the paper will meet the publication criteria if the following comments can be well responded.

1. There are two types of search functions when using Google Trends: “search term” and “topic”. They usually generate completely different results when using the same keyword. Please explicitly specify which one was used in Section 2.

2. People in non-English speaking countries are also likely to use English search terms, and the likelihood varies across countries. For example, I tested the term “relationship” versus “beziehung”+ “partnerschaft” in Germany and Austria during Jan to Jun 2020, “relationship” accounts for 22% and 24% of its German corresponding words in terms of Google search volumes. However, in France and Spain, it only accounts for 1.5% and 1.6% of its French and Spanish corresponding words. Such differences should not be neglected. Therefore, the authors should explain how do they reduce language bias.

3. I am not sure if the term “event study” is appropriate here. According to Fama, Fisher Jensen and Roll (1969) and Binder (1998), the event study methodology was initially proposed to examine the effect of a stock split announcement on stock prices. The method was later used in many accounting, economic and finance domains. However, the method does not seem to be used in many sociological studies. If the authors can justify it by referencing several seminal articles in sociology/demography used the same term, it would be fine to keep it. Otherwise, it is suggested to use a new phrase.

Reviewer #2: Summary

The paper leverages variability in lockdown timing within and between countries to identify effects of lockdowns on demography-relevant internet searches with potential implications for subsequent demographic trends.

Strength of the paper

The paper is well written and structured. It is also technically sound. The authors propose an index to compare search interests across countries, an interesting proposal to overcome the limitations of the Google data, which is pre-standardised. Furthermore, the authors conduct a thorough treatment of the Google data accounting from contextual factors. Crucially, they focus on the influence of outliers such as demographic events happening to celebrities, which affected search volumes and trends. The paper reports several robustness checks. The paper studies the phenomenon of interest using event study and diff-in-diff models convincingly.

Weaknesses of the paper

On the framing of the paper

The introductory section offer a succinct and pertinent overview of the expected effects of the pandemic on future demographic trends. However, the authors mention that we can only ‘hypothesize that the pandemic and related lockdowns will have an impact on future family demographic trends’. While this is true, there have been a number of recent studies that have provided evidence on things like fertility intentions, which are in themselves an outcome of interest for family demographers (https://www.demographic-research.org/volumes/vol43/47/). I understand that some of this literature might have been published after the paper was submitted, but acknowledging it will help the motivation of the paper stronger. Note that this is not to say that the literature review is outdated (e.g., the Wilde et al paper is very recent).

The authors focus on Europe and the US but do not justify their geographic focus. Given that Google data is, to some extent global, why were only these regions chosen? Up to the end of page four, I had the impression that the authors were giving a global picture of the spread of the pandemic and its consequences for family demography. In the last paragraph of page four, there is a sudden jump to a focus on Europe and the US: “As such, we expect considerable heterogeneity across countries—and, in particular, between Europe and the United States—in the degree to which individuals and families experience economic uncertainty and hardship.” The authors should explain why we would expect particularly high levels of heterogeneity between Europe and the United States? Can we really expect them to vary more compared to other regions of the Global South?

The assertion that recent “Google search trends are useful for predicting subsequent demographic trends” should be qualified. Wilde et al. (2020) create predictions about future fertility but whether these will turn out to be true or not is an open question at the time of writing. Furthermore, the authors show no evidence of search data actually, and successfully, predicting any demographic trends. This limits the believability of the claim that the claim that web searches are “likely to provide insight into future trends in [demographic] domains”. I suggest including references to nowcasting demographic dynamics using social media data as a base for these claims. The work by Zagheni et al. is particularly useful for this (e.g., https://www.jstor.org/stable/26622775?seq=1#metadata_info_tab_contents) and Billari, who you do cite elsewhere.

On measurement and analysis:

In page 5, Data section, the authors mention that the list of search terms is a crucial component of the study. The authors say that they “identified a series of search terms for each construct”. However, it is not clear how these terms were chosen and why they vary so much across languages. The text says that the “translations focused on achieving consistent cultural, linguistic, and in-country meaning rather than direct translation of specific words” but it would be good to know the actual mechanics of how the terms were chose – were in-country experts consulted? Was the initial keywords list taken from a published study? Considering the central importance of keyword choice it is somewhat surprising that the authors do not provide, as a sensitivity analysis, results for alternative keyword configurations. Do the results change if some keywords are incuded/excluded? Does it matter that some languages have different number of keywords for each domain (e.g. one word for ‘abortion’ in English and four in German)? The Mexican Spanish “pastilla abortive” is incorrect (it should be “abortiva”).

The authors state that “In order to observe the potential impact of lockdown policies on search trends, we also collected country- and state-specific lockdown dates2 as well as data on COVID-19 case, death, and recovery rates”. It would be good to know the source of these data and their granularity (e.g. weekly, sub-national). This is important because, in page 10, the authors mention that they use these data to calibrate their models. E.g., “First, we controlled for the number of confirmed COVID cases or the number of COVID related-deaths”. From a demographic perspective, it would be interesting to know why the authors use Covid cases/deaths as an indicator of pandemic lethality, rather than the more commonly-used excess death measure, which is available on a weekly basis from the HMD STMF: https://www.mortality.org/. Similarly, it is unclear if the authors used weekly data on Covid cases/deaths. One is led to believe so given statements like: “Controlling for the number of cases tended to reduce the magnitude of the estimates. However, this also reflects that the number of confirmed cases was higher in the period closely proceeding lockdowns and in the subsequent weeks” but this should be explicit.

Table 2 in the Appendix has lockdown dates, which presumably are the dates of the start of lockdowns. There are no dates for the lifting or changing of the policies on restrictions on movement. Countries introduced different restrictions in the time considered, relaxing and constraining measures over time. Similarly, lockdowns in Europe became more localised over time (i.e., going from national- to regional-level). I believe that the Oxford COVID-19 Government Response Tracker has been keeping track of these data but confess that I have not used these data personally. This has implications for the statistical analyses as well: it seems that lockdowns are treated as binary outcomes in model 1 (i.e., t indicates time to/from lockdown start). The same applies for the diff-in-diff model. You state that “we now include a single binary variable which is coded “1” during the period in which a lockdown is in effect, denoted as 1{ ≥ 0}, rather than the indicators for weeks preceding of following a lockdown.” Does this account for the changing nature of lockdowns over time? If so, it would be could to make this explicit. Please clarify how, and if, you account for these nuanced in your measurement of lockdowns. If you do not account for this, please also mention it as a limitation of the research.

I am generally convinced by the standardised index of search interest proposed by the authors. However, I would appreciate it if the authors could elaborate on what they mean by “It can be shown that ~ = ®̅̅ , where ® is the average index of relative searches for a term in country c over the January 2016-June 2020 period.” There is no need to include this in the text, I just want to make sure that I understand what they mean by this last statement.

It is unclear how search outliers were identified. From the discussion in page 9, it appears that outliers were deduced from the data (e.g., Fig 1 in Appendix). However, there were presumably many ‘smaller’ outliers that were not so evident and might have biased the data (e.g., subnational scandals or events not in English). The last paragraph in page 10 suggest that this might have been the case. The last robustness check was, according to the authors, skewed by an (presumably not previously identified) abortion-related debate associated with the “Human Life Protection Act” in the US. Can we expect similar mini-scandals to also skew Google Trends data systematically? Was there any systematic attempt to identify potential outliers using other data sources?

About the Results

In Table 1, why is the number of observations only reported for some regions (e.g., not included for France)? It would be useful to know how you treated missing values (i.e., country/week combinations for which Google returned no results). In the footnote of Table 1 for example, you mention that “Spanish searchers in the U.S. returned null searches in all states except California, Florida, New York, and Texas.”. However, the table provides betas for all domains, except for Emergency pill and Plan Child. Can you please explain the logic behind this and how to interpret these coefficients given the high number of missing values?

In the introduction, the authors hypothesize that lockdowns will lead to fewer demographic consequences in countries that better managed the economic fallout from the pandemic and provided a stronger financial safety net. However, the paper never returns to this hypothesis. If it is central to the study, it should be discussed. If it is not, the authors may want to reconsider whether it should be stated at all.

The authors need to clarify the mechanisms linking search behaviour to future fertility. The paper assumes that changes in search behaviour may be an indicator of future fertility behaviour but do not, in my opinion, provide ant credible evidence to that respect. Admittedly, doing so is without the scope of this paper, but the authors could refer to other studies that have indeed established a robust link between, for example, searches for ‘family planning’ and fertility nine months later in other contexts (ie, pre-covid). Otherwise, claims like: “Notably, however, we do observe a significant 18 percent increase in such searches in France in the post-lockdown period. This may mean that lockdown will have positive effects on fertility in France, which has particularly generous family policies” are less credible than they could be. The same is true of the concluding statement that “Overall, our results do not suggest that there will be a broad impact of national lockdowns on future family demography in European countries”.

Related to the last point, I would suggest that the authors do not generalise their results beyond the observed evidence. This applies to the conclusion that: “In sum, using Google Trends data, we find evidence that pandemic-induced lockdowns may have small and likely short-lived impacts on future family formation and dissolution behaviours, particularly in the U.S., but are unlikely to influence subsequent fertility.” The authors acknowledge in page 18 that “Google searches do not directly measure behaviors. Future research based on actual levels of demographic behaviors is needed to corroborate whether and how such searches may predict trends in family demography.” As a result, it seems unusual to be so conclusive regarding future demographic behaviour extrapolated from search terms alone.

Other comments:

- The Reference section does not list all cited paper. Please review this and make sure that all sources are included; e.g., Perrotta et al. (2020).

- I suggest proof-reading the document to remove spelling errors and typos, e.g.,

o “Compared to Europe, returns to average search levels were less evident for the U.S. where, even 2 to 3 months after lockdowns were introduced”

o “Google Trends provides data the relative search index = / only”)

o “Spanish searchers in the U.S. returned null searches in all states except California, Florida, New York, and Texas.. “

Reviewer #3: This study uses a valuable data source – Google Trends – to study the impact of the COVID-19 pandemic on union formation/dissolution and fertility. One limitation of Google Trends is that we cannot account individuals’ search intentions (e.g. are individuals searching for ‘wedding’ because they are planning a wedding, or because they are curious about how the pandemic has impacted weddings?). However, Google Trends does capture, given what we know about internet use rates from the World Bank [1] and the popularity of Google as a search engine [2], a broad swath of the population. It is also a source of longitudinal data, and is thus an excellent tool for monitoring attitudes pre- or post-exogenous shock.

Perhaps anticipating any backlash regarding the non-representativeness of digital data, the authors handle Google Trends data delicately and provide methodological context critical for interpreting results. This study is among the first I have read to not only offer a through description of what the Google Trends index represents, but explicitly pair this description with the study’s objectives. I appreciate that the authors account for non-pandemic events that may skew search volume, though I would appreciate additional detail on how this identification process occurred and whether it is replicable by other researchers. A systematic, generalizable approach to cleaning search query data could be of great use to other researchers.

A number of the authors’ findings have face validity when framed in the context of an acute social crisis. However, the fact that searches for divorce increase slightly immediately after lockdown and beyond is surprising, and does not correlate with my expectations given suggestions about economic uncertainty and union dissolution. To me, this unusual finding highlights this study’s need for explicit hypothesis identification at the onset. The authors’ introduction provides a thorough analysis of literature regarding union formation/dissolution and fertility during times of crisis, and the discussion provides anticipated justification for observed trends. However, because there are so many moving parts associated with this study I would encourage the authors to explicitly state a.) what associations they plan to analyze and b.) precisely how these relationships are expected to behave given contextual factors such as location or outbreak severity prior to jumping into their analysis.

I would also encourage the authors to be careful about framing this analysis in relation to past analyses of exogenous shocks on fertility and union formation. They suggest that overall interest in union formation/dissolution and fertility has not consistently changed across contexts, and may not change in the future. However, I think this is a premature statement given the widespread economic impact of the lockdown in areas with limited government support, and I would refrain from speculating until post-event data is available.

One concern I have with this paper lies in the ambiguity of the term “lockdown,” and selection of a post-lockdown period across contexts. A number of contexts, including many states in the United States, have consistently upheld public health related restrictions that would reasonably inhibit individuals’ ability to leave their homes and/or interact with others. Not only would the duration of these quasi-lockdown states have the potential to extend post-event drops in interest regarding acute fertility concerns (e.g. condoms, abortion, etc.), they could result in greater economic downturns and a lack of consumer confidence, thus shaping demographic trends for years to come. The difference in search volume for relationship/dating terms post-lockdown in Europe versus the United States is a good example of how variability in lockdown measures may impact choices, as travel and recreation remains severely limited in many states well into the latter half of 2020. A more explicit definition of what constitutes lockdown measures may help unpack this trend.

Finally, while the authors state that they interpret observed trends within the context of social conditions and economic hardship across contexts, it may be helpful to identify and control for economic impact within their models. As existing literature on fertility and union formation suggests, economic stability is an essential driver in the decision to couple/separate and/or have children. While we have yet to observe the long-term macroeconomic impact of the pandemic, there may be ways to measure perceived economic uncertainty – such as job losses or gains per month - that could provide insight into observed trends.

Overall, this is a fascinating and timely paper that hints at the potential for long-term demographic shifts post-pandemic. With some reframing and analytic adjustments, I believe it could be of great interest to the PLOS readership.

[1]https://data.worldbank.org/indicator/IT.NET.USER.ZS

[2]https://gs.statcounter.com/search-engine-market-share

6. PLOS authors have the option to publish the peer review history of their article (what does this mean?). If published, this will include your full peer review and any attached files.

Reviewer #1: No

Reviewer #2: **Yes: **Diego Alburez-Gutierrez

Reviewer #3: No

---

## [Author Response · Author response to Decision Letter 0]

4 Feb 2021

We attached a file with our responses to each of the three reviewers.

---

## [Decision Letter · Decision Letter 1]

19 Feb 2021

COVID-19 Lockdowns and Demographically-Relevant Google Trends: A Cross-National Analysis

PONE-D-20-38713R1

Dear Dr. Solaz,

We’re pleased to inform you that your manuscript has been judged scientifically suitable for publication and will be formally accepted for publication once it meets all outstanding technical requirements.

You can take reviewer 3 comments as suggestions that might help in further work and refinement of the methods proposed.

Kind regards,

José Antonio Ortega, Ph.D.

Academic Editor

PLOS ONE

Additional Editor Comments (optional):

Reviewers' comments:

Reviewer's Responses to Questions

**Comments to the Author**

1. If the authors have adequately addressed your comments raised in a previous round of review and you feel that this manuscript is now acceptable for publication, you may indicate that here to bypass the “Comments to the Author” section, enter your conflict of interest statement in the “Confidential to Editor” section, and submit your "Accept" recommendation.

Reviewer #1: All comments have been addressed

Reviewer #2: All comments have been addressed

Reviewer #3: (No Response)

2. Is the manuscript technically sound, and do the data support the conclusions?

Reviewer #1: Yes

Reviewer #2: Yes

Reviewer #3: Partly

3. Has the statistical analysis been performed appropriately and rigorously? 

Reviewer #1: Yes

Reviewer #2: I Don't Know

Reviewer #3: Yes

4. Have the authors made all data underlying the findings in their manuscript fully available?

Reviewer #1: Yes

Reviewer #2: Yes

Reviewer #3: Yes

5. Is the manuscript presented in an intelligible fashion and written in standard English?

Reviewer #1: Yes

Reviewer #2: Yes

Reviewer #3: Yes

6. Review Comments to the Author

Reviewer #1: The authors have considered the reviewer’s comments and responded well. The manuscript has been revised accordingly. Therefore, I recommend this paper for publication.

Reviewer #2: The authors have addressed all of my comments satisfactorily. The new robustness checks are a good addition to the manuscript.

Reviewer #3: The authors have made changes to the discussion of this manuscript that address my concerns regarding the ambiguity of the term “lockdown” and what this means for their finding. Lockdowns do vary in intensity, and it may be difficult to quantify this restriction. I still have reservations about lockdown severity and compliance, but this may lie beyond this scope of this study.

I did not note any significant changes to the discussion of outlier identification. Overall, the discussion of term identification and data cleaning does not seem replicable for other researchers interested in exploring this topic. If replicability is a challenge in the context of analyses that use search data, I would encourage the authors to be transparent about this.

My primary concern with the first iteration of this study concerned how the authors established hypotheses and clarified links between search behavior, fertility/family planning concerns, economic uncertainty and lockdown restrictions. I have similar concerns with this iteration of the paper. The introduction cites that lockdown may result in increased union dissolution as couples deal with increased stress, or it may result in more relationship stability as couples spend more time together. Fertility may increase as the couples have more time to focus on their families and relationship. However, the uncertainty of the pandemic may also reduce fertility rates. The economic fallout of the pandemic and the strictness of lockdown and physical distancing mandate are expected to serve as an important mediating factors that may produce variation in search results across contexts. Overall, it is clear that there are a number of factors that may shape fertility and union concerns, and that we can identify at least a couple of mechanisms underlying anticipated changes.

The results of this study focus primarily on whether demographically relevant keyword searches increase or decrease across countries. What seems to be missing are consistent links between the proposed mechanisms shaping fertility and union formation/dissolution concerns across contexts, anticipated changes in search volume across country context, and confirmation or refutation of anticipated search trends. The authors may consider adding refining their hypotheses by classifying countries studied according to the severity of lockdown restrictions and economic impact they experienced, and establishing anticipated fertility and/or union dissolution trends for each economic uncertainty and/or lockdown intensity category. While it is difficult to fully anticipate how these factors may impact demographic choices given the influence of additional potential control factors and individual agency, explicitly structured hypotheses may help link together sections of this study.

The authors have already taken a step toward this by controlling for unemployment in their models. In their conclusion, they state: “[we] hypothesized that consequences for demographic behaviors might be more marked in countries with less well managed outbreaks and with fewer safety nets, exacerbating feelings of uncertainty and economic insecurity.” They indeed find interesting patterns regarding searches for divorce across contexts, when controlling for unemployment search volume. Accounting for this potential mechanism provides an interesting story, and allows us to do more than simply observe how possible fertility and union formation/dissolution interest may have increased or decreased at the start of the pandemic.

Overall, this study illustrates excellent use of a novel data source, and identifies very interesting patterns in demographically relevant searches pre and post-lockdown. However, I think more work needs to be done to render observed patterns interpretable and grounded in anticipated mechanisms known to drive fertility and union dissolution trends. Accounting for unemployment is a strong start in this direction, but establishing more straightforward, mechanism-focused hypotheses may help.

7. PLOS authors have the option to publish the peer review history of their article (what does this mean?). If published, this will include your full peer review and any attached files.

Reviewer #1: No

Reviewer #2: **Yes: **Diego Alburez-Gutierrez

Reviewer #3: No

---

## [Editor Report · Acceptance letter]

3 Mar 2021

PONE-D-20-38713R1 

COVID-19 Lockdowns and Demographically-Relevant Google Trends:A Cross-National Analysis 

Dear Dr. Solaz:

I'm pleased to inform you that your manuscript has been deemed suitable for publication in PLOS ONE. Congratulations! Your manuscript is now with our production department. 

Kind regards, 

on behalf of

Dr. José Antonio Ortega 

Academic Editor

PLOS ONE